# MemoNav: Working Memory Model for Visual Navigation

## Abstract

We present MemoNav, a novel memory model for image-goal navigation, which utilizes a working memory-inspired pipeline to improve navigation performance. Specifically, the node features on the topological map are stored in the short-term memory (STM), as these features are dynamically updated. The MemoNav retains the informative fraction of the STM via a forgetting module to improve navigation efficiency. To learn a global representation of 3D scenes, we introduce long-term memory (LTM) that continuously aggregates the STM. Afterward, a graph attention module encodes the retained STM and the LTM to generate working memory (WM). After encoding, the WM contains the informative features in the retained STM and the scene-level feature in the LTM and is finally used to generate actions. Consequently, the synergy of these three types of memory increases navigation performance by selectively retaining goal-relevant information and learning a high-level scene feature. When evaluated on multi-goal tasks, the MemoNav outperforms the SoTA methods at all difficulty levels in both Gibson and Matterport3D scenes. The MemoNav also achieves consistent improvements on traditional 1-goal tasks. Moreover, the qualitative results show that our model is less likely to be trapped in a deadlock.

## 1 Introduction

This paper studies the image-goal navigation (ImageNav) problem, which aims to steer an agent towards a destination with the goal image in unseen environments. ImageNav has recently received much attention due to its promising applications in robots navigating in the open world.

Scene memory is essential for imageNav as it provides indispensable historical information for decision-making in unseen environments (Savinov et al., 2018). During navigation, this memory typically stores both scene features and the agent's navigation history (Kwon et al., 2021). These two types of information in turn help the agent generate more reasonable navigation actions by lessening the negative impact of partial observability (Parisotto & Salakhutdinov, 2018). In literature, various memory mechanisms have been introduced for ImageNav, which can be classified into three categories according to memory structure: (a) metric map-based methods (Chaplot et al., 2020a; Chen et al., 2019) that reconstruct local top-down maps and aggregate them into a global map, (b) stacked memory-based methods (Pashevich et al., 2021; Mezghani et al., 2021; Fang et al., 2019) that stack the past observations in chronological order, and (c) topological map-based methods (Kwon et al., 2021; Chaplot et al., 2020b; Beeching et al., 2020; Savinov et al., 2018) that store sparse landmark features in graph nodes. The topological map-based methods benefit from the memory sparsity of topological maps and have achieved impressive performance in ImageNav.

However, existing topological map-based methods still suffer from two major limitations: (a) Unawareness of useful nodes. They generally use all node features for generating actions without considering the contribution of each node, thus being easily misled by redundant nodes that are uninformative of the goal. (b) Local representation. Each node feature only represents a small local area in a large scene, which restricts the agent's ability to learn a higher-level semantic and geometric representation of the scene.

To overcome the above two limitations, we present a novel ImageNav method named MemoNav, which is motivated by the classical concept of working memory in cognitive neuroscience (Cowan, 2008) and in loose analogy with the working memory model in human navigation (Blacker et al.,

2017). The MemoNav learns three types of scene representations: (a) Short-term memory (STM) represents the local and transient features of the nodes in a topological map. (b) Long-term memory (LTM) is a global node that learns a scene-level representation by continuously aggregating STM. (c) Working memory (WM) learns goal-relevant features about 3D scenes and is used by a policy network to generate actions. The WM is formed by encoding the informative fraction of the STM and the LTM.

Based on the above three representations, the MemoNav navigation pipeline contains five steps: (1) STM generation. The map update module stores landmark features on the map as the STM. (2) Selective forgetting. To incorporate goal-relevant STM into the WM, a forgetting module temporarily removes nodes whose attention scores assigned by a memory decoder rank below a predefined percentage. After forgetting, the navigation pipeline will not compute the forgotten node features at subsequent time steps. (3) LTM generation. To assist the STM, we add a global node to the map as the LTM. The global node links to all map nodes and continuously aggregates the features of these nodes at each time step. (4) WM generation. A graph attention module encodes the retained STM and the LTM to generate the WM. The WM utilizes the goal-relevant information in the retained STM and the scene-level feature in the LTM, thus enabling the agent to utilize informative scene representations to improve navigation performance. (5) Action generation. Two Transformer decoders use the embeddings of the goal image and the current observation to decode the WM. Then, the decoded features are used to generate navigation actions.

Consequently, with the synergy of the three representations, the MemoNav noticeably outperforms the SoTA method (Kwon et al., 2021) in the Gibson scenes (Xia et al., 2018), increasing the navigation success rate by approximately 2.9%, 1.4%, 2.4%, and 1.7% on 1, 2, 3, and 4-goal test datasets, respectively. The comparison in the Matterport3D scenes (Chang et al., 2017) shows that the MemoNav exhibits better transferability.

The main contributions of this paper are as follows:

- We propose the MemoNav, which learns three types of scene representations (STM, LTM, and WM) to improve navigation performance in the ImageNav task.

- We use a forgetting module to retain the STM, thereby reducing redundancy in the map and improving navigation efficiency. We also introduce a global node as the LTM. The LTM connects to all nodes in the STM and learns a scene-level representation that provides the agent with a global view.

- We adopt a graph attention module to generate WM from the retained STM and the LTM. This module flexibly adjusts weights used for aggregating node features, which helps the agent use adaptive WM to improve performance.

- The experimental results demonstrate that our model outperforms the SoTA baseline on both 1-goal and multi-goal tasks in two popular scene datasets.

## 2 RELATED WORK

**ImageNav methods**. Since an early attempt (Zhu et al., 2017) to train agents in a simulator for ImageNav, rapid progress has been made on this task (Beeching et al., 2020; Chen et al., 2021; Wasserman et al., 2022; Al-Halah et al., 2022). Several methods have utilized topological scene representations for visual navigation, of which SPTM (Savinov et al., 2018) is an early work. NTS (Chaplot et al., 2020b) and VGM (Kwon et al., 2021) incrementally build a topological map during navigation and generalize to unseen environments without exploring the scenes in advance. These methods utilize all features in the map, while the MemoNav flexibly utilizes the informative fraction of these features. Another line of work (Yadav et al., 2022; Majumdar et al., 2022) has introduced self-supervised learning to enhance the scene representations, achieving a promising navigation success rate. In contrast, we enhance the scene representations using a global node that aggregates the agent's local observation features.

**Memory mechanisms for reinforcement learning**. Several studies (Ritter et al., 2021; Lampinen et al., 2021; Sukhbaatar et al., 2021; Loynd et al., 2020) draw inspiration from memory mechanisms of the human brain and design reinforcement learning models for reasoning over long time horizons. Ritter et al. (Ritter et al., 2021) proposed an episodic memory storing state transitions for navigation

Figure 1: **Example episodes for 1-goal and three-goal tasks.** In the 1-goal task (left), the agent searches for the goal during exploring the scene. In the three-goal task (right), the agent continues to navigate to two more successive goals.

tasks. Lampinen et al. (Lampinen et al., 2021) presented hierarchical attention memory as a form of "mental time-travel" (Tulving, 1985), which means recovering goal-oriented information from past experiences. Unlike this method, our model retains such information via a novel forgetting module. Expire-span (Sukhbaatar et al., 2021) predicts life spans for each memory fragment and permanently deletes expired ones. Our model is different from it in that we restore forgotten memory if the agent returns to visited places. The most similar work is Working Memory Graph (Loynd et al., 2020) which is also inspired by working memory in cognitive neuroscience. However, its memory capacity is fixed. In contrast, our model retains a certain proportion of short-term memory in the working memory and adjusts the memory capacity when the navigation episode contains more goals.

## 3 BACKGROUND

### 3.1 TASK DEFINITION

The objective of ImageNav (Figure 1) is to learn a policy $\pi$ to reach a target, given an image $\mathbf{I}_{target}$ that contains a view of the target and a series of observations $\{\mathbf{I}_t\}$ captured during the navigation. At the beginning of navigation, the agent receives an RGB image $\mathbf{I}_{target}$ of the target. At each time step, the agent captures an RGB-D panoramic image $\mathbf{I}_t$ of the current location and generates a navigational action. Following (Kwon et al., 2021), any additional sensory data (e.g. GPS and IMU) are not available.

### 3.2 BRIEF REVIEW OF VISUAL GRAPH MEMORY

The MemoNav is mainly built upon VGM (Kwon et al., 2021), which is briefly introduced below.

VGM incrementally builds a topological map $\mathcal{G} = (\mathcal{V}, \mathcal{E})$ from the agent's past observations where $\mathcal{V}$ and $\mathcal{E}$ denote nodes and edges, respectively. The node features (short-term memory) $\boldsymbol{V} \in \mathbb{R}^{d \times N_t}$ are generated from observations by a pretrained encoder $\mathcal{F}_{loc}$ where $d$ denotes the dimension of feature and $N_t$ the number of nodes at time $t$.

VGM uses a graph convolutional network (GCN) to encode the topological map, obtaining the encoded memory $\boldsymbol{M} = \text{GCN}(\boldsymbol{V})$. Before encoding, VGM obtains the target embedding $\boldsymbol{e}_{target} = \mathcal{F}_{enc}(\mathbf{I}_{target})$ and fuses each node feature with this embedding using a linear layer.

The encoded memory is then decoded by two Transformer (Vaswani et al., 2017) decoders, $\mathcal{D}_{cur}$ and $\mathcal{D}_{target}$. $\mathcal{D}_{cur}$ takes the current observation embedding $\boldsymbol{e}_{cur} = \mathcal{F}_{enc}(\mathbf{I}_{cur})$ as the query and the feature vectors of the encoded memory $\boldsymbol{M}$ as the keys and values, generating a feature vector $\boldsymbol{f}_{cur}$. Similarly, $\mathcal{D}_{target}$ takes the target embedding $\boldsymbol{e}_{target}$ as the query and generates $\boldsymbol{f}_{target}$. Lastly, an LSTM-based policy network takes as input the concatenation of $\boldsymbol{f}_{cur}$, $\boldsymbol{f}_{target}$ and $\boldsymbol{e}_{cur}$ to output an action distribution.

## 4 METHOD

The MemoNav comprises three key components: the forgetting module, the global node, and the GATv2 memory encoder. We show the pipeline of the MemoNav in Figure 2 and describe these components in the remainder of this section.

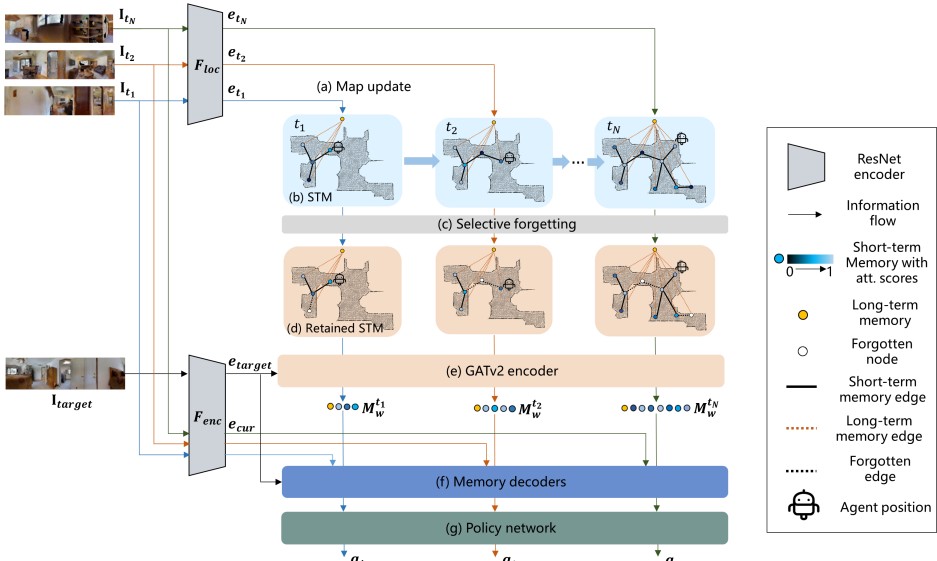

Figure 2: **Overview of MemoNav.** (a) The memory update module builds a topological map using $e_t$, the embedding of the current image $\mathbf{I}_t$. (b) The node features in the map form the STM while a global node that links to each node acts as the LTM. (c) The forgetting module temporarily excludes a fraction of STM whose attention scores rank below a threshold $p$. (d) The retained STM and the LTM are then encoded by (e) a graph attention module to generate the WM $M_w$. (f) The WM is decoded by two Transformer decoders (details in the Appendix). (g) Lastly, the output of the decoding process is input to a policy network to generate navigation actions.

## 4.1 SELECTIVE FORGETTING MODULE

We devise a forgetting module that instructs the agent to forget uninformative experiences. Here, "forgetting" means that nodes with attention scores lower than a threshold are temporarily excluded from the navigation pipeline. This means of forgetting via attention is evidenced by research (Fukuda & Vogel, 2009) revealing that the optimal performance of working memory relies on humans' ability to focus attention on task-relevant information. In Figure 3, we visualize the attention scores for the short-term memory (STM) calculated in $\mathcal{D}_{target}$ of VGM. The figure shows that the agent assigns high scores to nodes close to targets while paying little attention to remote ones. This phenomenon indicates that not all node features in the STM are helpful and that it is more efficient to use a small number of them for navigation.

The forgetting module retains a fraction of STM according to the attention scores $\{\alpha_i\}_{i=1}^{N_t}$ in $\mathcal{D}_{target}$. These scores denote to what extent the goal embedding $e_{target}$ attends to each node feature in the STM. After $\mathcal{D}_{target}$ finishes decoding, the agent temporarily "forgets" a fraction of nodes whose scores rank below a predefined percentage $p$. In other words, these nodes will be disconnected from their neighbors, and not be processed by the navigation pipeline at the subsequent time steps. If the agent returns to a forgotten node, this node will be added to the map and processed by the pipeline again. In a multi-goal task, once the agent has reached a target, all forgotten nodes will be restored, as these nodes are probably useful for finding the next target.

The forgetting module is used in a plug-and-play manner, which means that this module is not activated during training and switched on when evaluating and deploying. $p$ is default to 20%. With this module, the agent can selectively retain the informative STM, while avoiding misleading experiences.

## 4.2 LONG-TERM MEMORY GENERATION

In addition to STM, the information in the long-term memory (LTM) also forms part of working memory (WM) (Ericsson & Kintsch, 1995). Inspired by ETC (Ainslie et al., 2020) and LongFormer

Figure 3: **VGM calculates redundant node features**. The figure visualizes the attention scores in $\mathcal{D}_{target}$ for an multi-goal episode of VGM. We can see that only a small number of nodes are useful for finding a single goal.

(Beltagy et al., 2020), we add a zero-initialized trainable global node $\boldsymbol{n}_{global} \in \mathbb{R}^d$ to the topological map as the LTM (the orange nodes in Figure 2). RecBERT (Hong et al., 2021) adopts a recurrent state token similar to the proposed LTM, but their functions are different. The token in RecBERT aggregates visual-linguistic clues while the LTM in our model summarizes the agent's past observations by continuously fusing the STM via memory encoding (the encoder is described in the next subsection).

The LTM possesses two advantages: it learns a scene-level feature and facilitates feature fusing. A recent study (Ramakrishnan et al., 2022) suggests that embodied agents need to learn higher-level representations of the 3D environment to overcome the partial observability caused by limited field-of-view sensors. From this viewpoint, the LTM stores a high-level scene representation by aggregating local node features. Moreover, another merit of the LTM is facilitating feature fusing. The topological map is divided into several sub-graphs when forgotten nodes are removed. Consequently, direct message passing between the nodes separated in different sub-graphs no longer exists. The LTM links to every node in the map, acting as a bypath that facilitates feature fusing between these isolated sub-graphs.

### 4.3 Working Memory Generation

The third type of scene representation WM learns goal-relevant features that are used to generate actions. To learn adaptive WM, we utilize a graph attention module GATv2 (Brody et al., 2022) to encode the retained STM and the LTM since GATv2 is powerful when different nodes have different rankings of neighbors. GATv2 derives adaptive weights from node features for neighboring nodes, instead of from the Laplacian matrix. This design is suitable for generating WM, especially in multi-goal tasks since the STM features that contain information about the target or a path leading to the target should obtain high weights while those of irrelevant places should receive lower weights. In addition, the node features in the STM are dynamically updated during navigation, as the agent replaces these features with new ones when revisiting these nodes. Therefore, it is more suitable to adaptively change the weights used to aggregate the STM features into the WM.

The WM generation is formulated as: $\boldsymbol{M}_w = \{\boldsymbol{V}', \boldsymbol{n}_{global}^t\} = \text{GATv2}(\{\boldsymbol{V}, \boldsymbol{n}_{global}^{t-1}\})$ where $\boldsymbol{M}_w$ represents the generated working memory, and $\boldsymbol{V}'$ the encoded STM. $\{\cdot, \cdot\}$ denotes that the LTM (a vector) is prepended to the retained STM (a sequence of vectors). Note that the time step superscript of $\boldsymbol{n}_{global}$ means that the LTM is recurrent through time.

After GATv2 encoding, the WM aggregates the goal-relevant information in the retained STM as well as the scene-level representation in the LTM. Lastly, the decoders $\mathcal{D}_{cur}$ and $\mathcal{D}_{target}$ take $\boldsymbol{M}_w$ as keys and values, generating $\boldsymbol{f}_{cur}$ and $\boldsymbol{f}_{target}$, which are further used to generate actions.

## 5 Experiments

### 5.1 Datasets

All experiments are conducted in the Habitat (Savva et al., 2019) simulator with the Gibson (Xia et al., 2018) or Matterpot3D (Chang et al., 2017) scene dataset. We adopt the same action space as in VGM (Kwon et al., 2021).

**1-goal dataset**. In the Gibson scenes, all models are trained with 72 scenes and evaluated on a public 1-goal [1] dataset (Mezghani et al., 2021) that comprises 14 unseen scenes. Following the setting in

---

[1] The 1-goal difficulty level here denotes the hard level in the public test dataset

VGM (Kwon et al., 2021), 1007 out of 1400 episodes in this public dataset are used for the evaluation on 1-goal tasks, but the 1400 episodes are still used for ablation studies.

**Multi-goal dataset**. Multi-goal evaluation is more suitable for evaluating memory mechanisms used by navigation methods. By letting the agent return to visited places we can test whether memory mechanisms help the agent plan a short path. If not, the agent will probably waste its time re-exploring the scene or traveling randomly. However, recent ImageNav methods seldom conduct multi-goal evaluations. To further investigate how memory mechanisms assist the agent in navigation, we follow MultiON (Wani et al., 2020) to collect 700-episode multi-goal test datasets in the Gibson scenes (an example in Figure 1). In a multi-goal task, the agent navigates to an ordered sequence of goals.

We follow five rules to set sequential goals for each sample: (1) No obstacles appear inside a circle with a radius of 0.75 meters centered at each goal. (2) The distance between two successive goals is no more than 10 meters. (3) All goals are placed on the same layer without altitude differences. (4) All goals are reachable from each other. (5) The final goal is placed near a certain previous one with the distance between them smaller than 1.5 meters. Please refer to A.4.4 for more details.

In the Matterpot3D scenes, we sample 1008 episodes for each difficulty level from the multi-goal test datasets used in Multi-ON (Wani et al., 2020). The difficulty of an episode is indicated by the number of goals. All compared methods and our model are trained on the Gibson 1-goal dataset and tested on other difficulty levels.

**Evaluation Metrics**. In 1-goal tasks, the success rate (**SR**) and success weighted by path length (**SPL**) (Anderson et al., 2018) are used. In a multi-goal task, two metrics are borrowed from (Wani et al., 2020): The progress (**PR**) is the fraction of goals successfully reached, equal to the **SR** for 1-goal tasks; Progress weighted by path length (**PPL**) indicates navigation efficiency and is defined as

$$PPL = \frac{1}{E} \sum_{i=1}^{E} Progress_i \frac{l_i}{\max(p_i, l_i)},$$ where $E$ is the total number of test episodes, $l_i$ and $p_i$ are the shortest path distance to the final goal via midway ones, and the actual path length taken by the agent, respectively.

## 5.2 Compared Methods and Training details

We compare the MemoNav with previous methods utilizing various types of memory. **Random** is an agent that samples actions from a uniform distribution and navigates with oracle stopping. **CNNLSTM** (Zhu et al., 2017) uses no maps but a hidden vector as implicit memory. **ANS** (Chaplot et al., 2020a) is a metric map-based model adapted for ImageNav in the experiments of VGM (Kwon et al., 2021). **NTS** (Chaplot et al., 2020b) incrementally builds a topological map without pre-exploring and adopts a hierarchical navigation strategy. **VGM** (Kwon et al., 2021) (see Section 3.2) is the baseline the MemoNav is built on. **ZER** (Al-Halah et al., 2022) is used to handle multiple navigation subtasks. **SLING** (Wasserman et al., 2022) uses mode switch to finish last-mile navigation. **OVRL** (Yadav et al., 2022) introduces offline visual representation learning before training agents.

Following VGM (Kwon et al., 2021), for all methods except ANS and NTS, the agent is first trained using imitation learning, minimizing the negative log-likelihood of the ground-truth actions. Next, the agent is finetuned with proximal policy optimization (PPO) (Schulman et al., 2017) to improve the exploratory ability. The reward setting and auxiliary losses remain the same as in VGM. Each model is trained for five runs on a TITAN RTX GPU. The evaluation results for ANS and NTS are borrowed from the VGM paper (Kwon et al., 2021) [2]

## 5.3 Quantitative Results

**Comparison on Gibson**. Table 1 shows that the MemoNav outperforms all the baselines in SR on the four difficulty levels. CNNLSTM exhibits the poorest performance as its limited memory provides insufficient information about the goal area. The MemoNav outperforms the metric map-based method ANS which requires pre-built maps, enjoying a large improvement in SR and SPL.

---

[2]ANS is designed for exploration while NTS is not opensourced, so it is not straightforward to reproduce them for multi-goal tasks.

Table 1: **Evaluation results on the Gibson (G) and Matterport3D (M) test datasets**. We report the averages and standard deviations (in parentheses) by repeating experiments five times. (G*: evaluation on 4200 validation trajectories, **SR**: success rate (%), **SPL**: success weighted by path length (%)), **PR**: progress (%), **PPL**: progress weighted by path length (%))

| Scene | Methods | 1-goal | | 2-goal | | 3-goal | | 4-goal | |
|---|---|---|---|---|---|---|---|---|---|
| | | SR | SPL | PR | PPL | PR | PPL | PR | PPL |
| G | Random | $17.4_{(0.7)}$ | $5.0_{(0.2)}$ | $4.9_{(0.6)}$ | $0.8_{(0.1)}$ | $3.1_{(0.3)}$ | $0.4_{(0.1)}$ | $1.8_{(0.2)}$ | $0.2_{(0.0)}$ |
| | CNNLSTM | $21.6_{(1.2)}$ | $14.6_{(1.1)}$ | $14.5_{(1.7)}$ | $9.6_{(0.9)}$ | $9.0_{(1.1)}$ | $2.8_{(0.2)}$ | $6.4_{(0.9)}$ | $1.6_{(0.1)}$ |
| | ANS | $30.0_{(-)}$ | $11.0_{(-)}$ | - | - | - | - | - | - |
| | NTS | $43.0_{(-)}$ | $26.0_{(-)}$ | - | - | - | - | - | - |
| | VGM | $71.8_{(1.7)}$ | $57.9_{(2.2)}$ | $45.6_{(2.9)}$ | $\mathbf{18.7}_{(1.5)}$ | $33.4_{(2.8)}$ | $8.3_{(0.8)}$ | $25.8_{(1.5)}$ | $5.0_{(0.4)}$ |
| | MemoNav (ours) | $\mathbf{74.7}_{(1.5)}$ | $\mathbf{58.1}_{(1.4)}$ | $\mathbf{47.0}_{(2.4)}$ | $18.3_{(1.1)}$ | $\mathbf{35.8}_{(2.9)}$ | $\mathbf{8.6}_{(0.4)}$ | $\mathbf{27.5}_{(1.0)}$ | $\mathbf{5.1}_{(0.2)}$ |
| G* | ZER (ViewAug) | 22.0 | 18.8 | - | - | - | - | - | - |
| | OVRL | 41.3 | 26.9 | - | - | - | - | - | - |
| | SLING+OVRLGD | 55.4 | $\mathbf{37.4}$ | - | - | - | - | - | - |
| | MemoNav (ours) | $\mathbf{62.2}$ | 36.6 | - | - | - | - | - | - |
| M | Random | $14.5_{(0.5)}$ | $5.5_{(0.3)}$ | $8.0_{(0.2)}$ | $0.9_{(0.0)}$ | $6.3_{(0.2)}$ | $0.5_{(0.0)}$ | - | - |
| | CNNLSTM | $16.2_{(1.3)}$ | $9.8_{(1.1)}$ | $10.8_{(0.9)}$ | $2.6_{(0.2)}$ | $7.7_{(0.8)}$ | $1.4_{(0.2)}$ | - | - |
| | VGM | $24.4_{(1.1)}$ | $\mathbf{16.3}_{(0.7)}$ | $15.8_{(1.7)}$ | $4.7_{(0.3)}$ | $12.0_{(0.2)}$ | $\mathbf{2.8}_{(0.2)}$ | - | - |
| | MemoNav (ours) | $\mathbf{26.0}_{(1.0)}$ | $16.1_{(0.4)}$ | $\mathbf{18.2}_{(0.7)}$ | $\mathbf{5.2}_{(0.2)}$ | $\mathbf{13.3}_{(0.2)}$ | $2.8_{(0.1)}$ | - | - |

Table 2: **Network component ablation results**. Row 1 is the baseline model VGM (Kwon et al., 2021), and row 7 is our full model. The averages over five runs are reported in this table while the standard deviations are placed in . (**G**: the GATv2-based memory encoder, **L**: the long-term memory, **F**: forgetting nodes whose attention scores rank below 20%)

| | Components | | | 1-goal | | 2-goal | | 3-goal | | 4-goal | |
|---|---|---|---|---|---|---|---|---|---|---|---|
| | G | L | F | SR | SPL | PR | PPL | PR | PPL | PR | PPL |
| 1 | | | | $55.8_{(3.3)}$ | $47.1_{(1.9)}$ | $45.6_{(2.9)}$ | $18.7_{(1.5)}$ | $33.4_{(2.8)}$ | $8.3_{(0.8)}$ | $25.8_{(2.7)}$ | $5.0_{(0.4)}$ |
| 2 | | ✓ | | $58.9_{(1.8)}$ | $48.4_{(1.2)}$ | $46.3_{(1.6)}$ | $18.3_{(1.0)}$ | $35.3_{(2.1)}$ | $8.3_{(0.9)}$ | $26.5_{(1.6)}$ | $4.7_{(0.5)}$ |
| 3 | | | ✓ | $56.8_{(3.2)}$ | $47.0_{(2.1)}$ | $45.8_{(2.9)}$ | $18.6_{(1.2)}$ | $33.7_{(2.9)}$ | $8.0_{(0.9)}$ | $25.9_{(3.3)}$ | $\mathbf{5.1}_{(0.5)}$ |
| 4 | | ✓ | ✓ | $59.4_{(1.7)}$ | $48.7_{(0.5)}$ | $46.3_{(1.9)}$ | $18.8_{(0.9)}$ | $35.4_{(1.1)}$ | $8.3_{(0.7)}$ | $26.8_{(1.3)}$ | $4.7_{(0.6)}$ |
| 5 | ✓ | ✓ | ✓ | $\mathbf{60.7}_{(1.9)}$ | $\mathbf{49.0}_{(1.0)}$ | $\mathbf{47.3}_{(2.4)}$ | $\mathbf{18.9}_{(1.1)}$ | $\mathbf{35.8}_{(1.4)}$ | $\mathbf{8.6}_{(0.4)}$ | $\mathbf{27.5}_{(1.0)}$ | $\mathbf{5.1}_{(0.2)}$ |

Compared with the VGM, our model exhibits a noticeable performance gain, increasing the SR by **2.9%**, **1.4%**. **2.4%**, and **1.7%** on the 1, 2, 3, and 4-goal tasks respectively while using 20% fewer node features. This result indicates that the MemoNav benefits from the informative scene memory and the high-level scene representation contained in the WM, obtaining a higher success rate.

The MemoNav is also tested using 90-degree FoV sensors instead of panoramas (see G* in Table 1. The evaluation results on 4200 trajectories (G*) used by ZER also show that the MemoNav outperforms ZER, OVRL, and SLING.

**Comparison on Matterport3D**. We conduct an evaluation on the Matterport3D scenes to test the models' ability to generalize to other scene types. Table 1 shows that our method achieves consistent performance improvements on this unseen scene dataset. Compared with VGM, the MemoNav witnesses performance gains in the success rate at the three difficulty levels, improving the SR/PR by **1.6%**, **2.4%**, and **1.3%** on the 1, 2, and 3-goal tasks, respectively. These results show that our method is more capable of generalizing to new scene styles.

## 5.4 ABLATION STUDIES AND ANALYSIS

We conduct ablation studies in the Gibson scenes to analyze the impact of each proposed component.

**Performance gain of each proposed component.** We ablate the three key components described in Section 4 and show the results in Table 2. Comparing row 2 with row 1, we can see that the LTM brings a noticeable improvement in SR/PR over the baseline. Comparing row 3 with row 1, applying the forgetting module achieves slight improvements in the SR/PR. However, its cooperation with the LTM witnesses larger increases (row 4 vs. row 1). More importantly, compared with the baseline (row 1), the synergy of the three components (row 5) increases the SR/PR by **0.049 (8.8%)**, **0.014 (3.1%)**,

Table 3: **Ablation study of LTM**. Row 1 is our default model. Row 2 shows the impact of replacing the LTM with a random feature in the STM. Row 3 is a variant that uses LTM to aggregates STM but does not incorporate the LTM into the WM. We report the averages and standard deviations (in parentheses) of five runs.

| | Methods | 1-goal | | 2-goal | | 3-goal | | 4-goal | |
|---|---|---|---|---|---|---|---|---|---|
| | | SR | SPL | PR | PPL | PR | PPL | PR | PPL |
| 1 | MemoNav | $60.7_{(2.1)}$ | $49.0_{(1.9)}$ | $47.0_{(2.4)}$ | $18.3_{(1.1)}$ | $35.8_{(1.4)}$ | $8.6_{(0.4)}$ | $27.5_{(1.0)}$ | $5.1_{(0.2)}$ |
| 2 | w/ random replace | $59.5_{(0.9)}$ | $47.1_{(1.1)}$ | $45.9_{(2.3)}$ | $17.3_{(1.3)}$ | $35.0_{(1.3)}$ | $6.7_{(0.4)}$ | $26.8_{(1.5)}$ | $4.1_{(0.3)}$ |
| 3 | w/o decoding LTM | $59.8_{(1.6)}$ | $48.4_{(1.3)}$ | $46.0_{(2.4)}$ | $17.5_{(1.0)}$ | $35.6_{(1.5)}$ | $7.9_{(0.3)}$ | $27.0_{(1.0)}$ | $4.8_{(0.4)}$ |

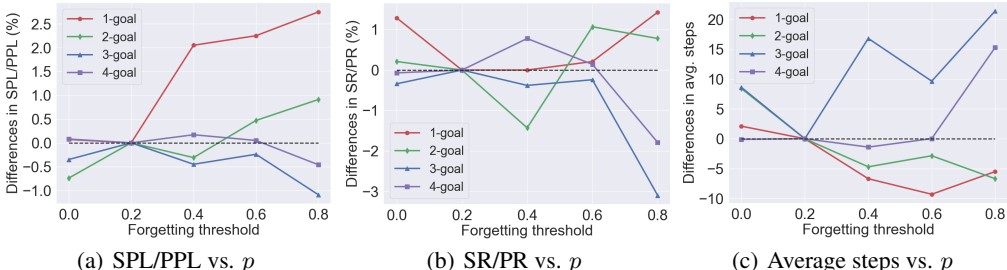

(a) SPL/PPL vs. $p$      (b) SR/PR vs. $p$      (c) Average steps vs. $p$

Figure 4: Navigation performance versus forgetting threshold $p$. (Averaged over five runs)

**0.024** (**7.2%**), **0.017** (**6.6%**) at the 1, 2, 3 and 4-goal levels respectively. These results demonstrate that the three components help to solve long-time navigation tasks with multiple sequential goals.

**Importance of LTM.** Table 3 presents the results of ablation experiments for the LTM (described in Section 4.2). The first ablation experiment (row 2) replaces the LTM feature with a randomly selected STM feature each time the GATv2 encoding is finished, so that the MemoNav cannot use the scene-level representation in the LTM. Row 2 shows that the navigation performance becomes worse at all difficulty levels compared with the full model (row 1). Furthermore, when the LTM feature is not incorporated into the WM (row 3), the performance also deteriorates. In summary, the LTM stores a scene-level feature essential for improving the success rate.

**Correlation between navigation performance and forgetting threshold**. We evaluate our model with different forgetting thresholds $p$ (see Section 4.1). The results are shown in Figure 4. For clarity, the figure shows the performance differences between our full model (with the forgetting module, the LTM, and the GATv2 encoder) and the variants. The four levels exhibit different trends. When $p$ increases (i.e. a larger fraction of STM is not incorporated into the WM), our model witnesses first increases and then drops in the SR/PR and SPL/PPL at the 3 and 4-goal levels while enjoying slight gains in these metrics at the 1 and 2-goal levels. At the 1-goal level, our model obtains the highest SR and SPL when $p = 80\%$, which means that a large fraction of node features in the map are useless when the navigation task is easy. A similar trend can also be seen at the 2-goal level. In contrast, our model exhibits an increase in the PR and PPL at the 4-goal level when $p$ rises from 0% to 40%. However, if $p$ rises to 80%, the two metrics see a precipitous decline, and the agent takes more than 20 steps to complete the tasks. These results suggest that excluding redundant STM from the WM improves navigation performance in multi-goal tasks. However, excluding an excessive fraction forces the agent not to utilize what it has explored, thus undermining the navigation performance.

## 5.5 VISUALIZATION RESULTS

To observe how the MemoNav improves the navigation performance, we show example episodes of CNNLSTM, VGM and our model in the Gibson scenes in Fig 5 (See Appendix for more). We can see that CNNLSTM takes numerous steps to explore and tends to fail in complex scenes. VGM tends to spend a large proportion of time going in circles in narrow pathways. In contrast, the trajectories of the MemoNav are shorter and smoother. For instance, the 2nd and 3rd columns in Figure 5 show

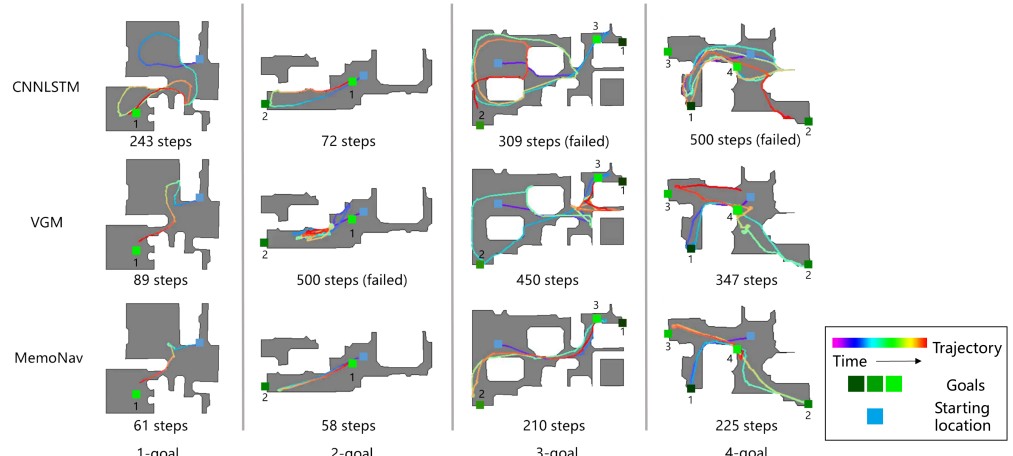

Figure 5: **Visualization of example episodes.** we compare selected episodes of CNNLSTM, VGM, and MemoNav at four difficulty levels in the Gibson scenes and visualize the top-down views. The number of navigation steps (the upper limit is 500) are shown at the bottom of each top-down view. Best viewed in color.

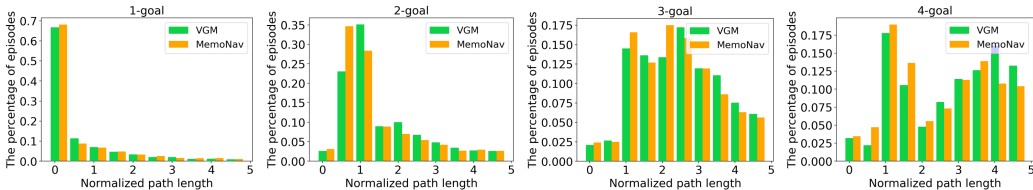

Figure 6: **Histograms of normalized path lengths of successful episodes for VGM (Kwon et al., 2021) and our model (averaged over 5 runs).** Our model has more short trajectories while VGM has a larger proportion of long episodes.

that the VGM agent is trapped in a bottleneck connecting two rooms, while our agent plans efficient routes. This comparison shows that the MemoNav is more capable of escaping from deadlock.

We next draw a histogram of the normalized path lengths of successful episodes for VGM and our model to further investigate to what degree our model escapes from deadlock. The normalized path length is calculated as $l_{norm} = l_i / \max(p_i, l_i) - 1$ where $p_i$ and $l_i$ are defined in Section 5.1. $l_{norm}$ indicates how many extra steps the agent takes compared to the shortest path. A larger $l_{norm}$ represents a less efficient navigation episode. The histogram is shown in Figure 6. The figure shows that the histogram of our model exhibits a less heavier-tailed distribution, especially at the 4-goal level. This distribution means that our model has fewer episodes with an extremely large number of steps and agrees with the qualitative comparison.

## 6 CONCLUSION AND FUTURE WORK

This paper proposes MemoNav, a novel memory model for ImageNav. This model flexibly retains the informative fraction of the short-term navigation memory via a forgetting module. We also introduce an extra global node as long-term memory to learn a scene-level representation. The retained short-term memory and the long-term memory are encoded by a graph attention module to generate the working memory that is used for generating action. The experimental results show that the MemoNav outperforms the baselines in multi-goal tasks, exhibits better transferability, and is more capable of escaping from deadlock. For future work, we will try to design trainable forgetting modules to better assess which fraction of the agent's memory is informative.

## REPRODUCIBILITY STATEMENT

We provide our source code and model configuration files in the supplementary material. We also add the python script used for generating our multi-goal Gibson datasets to the supplementary material. Any detail about data processing can be found in this script. For example, the rules used to generate multi-goal samples (described in Section 5.1) are coded in the script.

## LICENSES FOR REFERENCED DATASETS

Gibson: `http://svl.stanford.edu/gibson2/assets/GDS_agreement.pdf`

Matterprt3D: `http://kaldir.vc.in.tum.de/matterport/MP_TOS.pdf`

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

# A APPENDIX

## A.1 LIMITATIONS

While the MemoNav witnesses a large improvement in the navigation success rate in multi-goal navigation tasks, it still encounters limitations. The proposed forgetting module is a post-processing method, as it obtains the attention scores of the decoder before deciding which nodes are to be forgotten. Future work can explore trainable forgetting modules. The second limitation is that our forgetting module does not reduce memory footprint, since the features of the forgotten nodes still exist in the map for localization. Moreover, the forgetting threshold in our experiments is fixed. Future work can merge our idea with Expire-span (Sukhbaatar et al., 2021) to learn an adaptive forgetting threshold.

## A.2 POTENTIAL IMPACT

The notable potential of negative societal impact from this work: our model is trained on 3D scans of the Gibson scenes which only contain western styles. This inadequacy of diverse scene styles may render our model biased and incompatible with indoor environments in unseen styles. As a result, our model may be only available in a small fraction of real-life scenes. If our model is transferred to out-of-distribution scenes, the agent may take more steps and even bump on walls frequently.

## A.3 REPRESENTATIVE MODELS OF WORKING MEMORY IN HUMAN BRAIN

Cowan et al. (Cowan, 2008) proposed a typical model describing the relationships among long-term memory (LTM), short-term memory (STM), and working memory (WM) of the human brain. According to their definitions, LTM is a large knowledge base and a record of prior experience; STM reflects faculties of the human mind that hold a limited amount of information in a very accessible state temporarily; WM includes part of STM and other processing mechanisms that help to utilize STM. Cowan et al. designed a framework depicting how WM is formed from STM and LTM (shown in Figure 7). This framework demonstrates that STM is derived from a temporarily activated subset

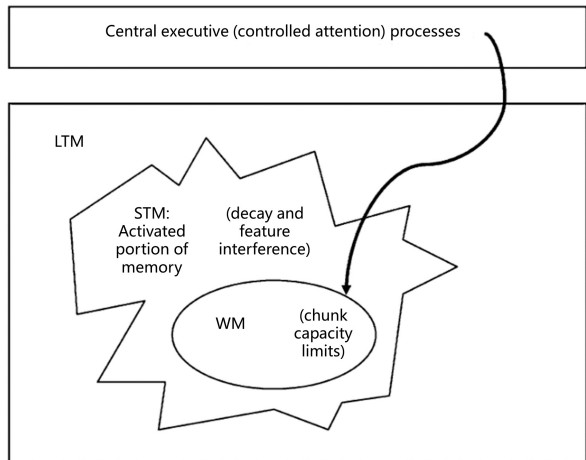

Figure 7: The memory model by Cowan et al. (Cowan, 2008). This figure is borrowed and adapted from its original paper.

of LTM. This activated subset may decay as a function of time unless it is refreshed. The useful fraction of STM is incorporated into WM via an attention mechanism to avoid misleading distractions. Subsequent work by Baddeley et al. (Baddeley, 2012) suggests that the central executive manipulates memory by incorporating not only part of STM but also part of LTM to assist in making a decision.

We draw inspiration from the work by Cowan et al. (Cowan, 2008) and Baddeley et al. (Baddeley, 2012) and reformulate the agent's navigation experience as the three types of memory defined above.

The parallel between the MemoNav and the two relevant models above is shown in the following list:

- The map node features are termed "STM", since they are local and transient.

- The topological map of the MemoNav maintains a 100-node queue to store map nodes. This design simulates STM that holds a limited amount of information in a very accessible state temporarily in the human brain.

- The MemoNav introduces a global node aggregating prior observation features stored in the topological map, thereby simulating LTM which acts as a large knowledge base.

- The MemoNav utilizes a forgetting mechanism to remove a fraction of STM with attention scores lower than a threshold. This mechanism acts as a simple way of decaying STM.

- The forgetting mechanism helps WM include part of STM.

- The MemoNav incorporates the retained STM and the LTM into WM, which is subsequently used to generate navigation actions. This design simulates the working memory model by (Baddeley, 2012).

## A.4 IMPLEMENTATION DETAILS

### A.4.1 IMPLEMENTATION OF MEMONAV

The structure of the memory decoding module (Figure 2(f) in the main paper) in the MemoNav remains the same as in the VGM (Kwon et al., 2021) and is shown in Figure 8. We maintain the module hyper-parameters specified in the supplementary of the VGM paper. The forgetting module on the MemoNav requires the attention scores generated in the decoder $\mathcal{D}_{target}$. Therefore, our model needs to calculate the whole navigation pipeline before deciding which fraction of the STM should be retained. This lag means that the retained STM is incorporated into the WM at the next time step. The pseudo-code of the MemoNav is shown in Algorithm 1

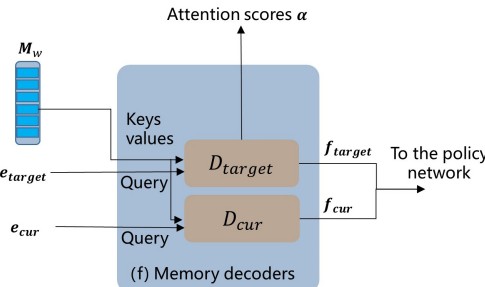

Figure 8: The structure of the memory decoding module displayed in Figure 2 (f) of the main paper. $M_w$ denotes the working memory. The attention scores $\alpha$ generated in $\mathcal{D}_{target}$ are used in the forgetting module to retain informative STM.

---

**Algorithm 1:** The implementation of the MemoNav

**Data:** Empty topological map $\mathcal{G} = \{\mathcal{V}, \mathcal{E}\}$, target image $\mathbf{l}_{target}$, current time step $t$, forgetting
percentage $p$, trainable observation encoder $\mathcal{F}_{enc}$, GATv2-based encoder GATv2,
Transformer decoders $\mathcal{D}_{target}$ and $\mathcal{D}_{cur}$, LSTM-based policy network LSTM

**Result:** Navigation action $a_t$

1  Long-term memory $\boldsymbol{n}_{global} \leftarrow \mathbf{0} \in \mathbb{R}^d$;
2  Attention scores for graph nodes $V$: $\alpha \leftarrow \mathbf{0} \in \mathbb{R}^{|V|}$;
3  **while** *not* AgentCallStop () **do**
  // Step 1:  Update the topological map
4   $\mathbf{l}_t \leftarrow$ GetCurrentPanorama();
5   $G.$UpdateMap($\mathbf{l}_t$);
  // Step 2:  Retain the informative fraction of the STM
6   Forgotten number $n \leftarrow$ Floor $(p \cdot |\mathcal{V}|)$;
7   Sorted indices $i \leftarrow$ Argsort($\alpha$);
8   Forgotten indices $i_{forgotten} \leftarrow i\,[0:n]$;
9   $G.$RemoveNodes($i_{forgotten}$);
  // Step 3:  Memory encoding and decoding
10  $\boldsymbol{V} \in \mathbb{R}^{|\mathcal{V}| \times d} \leftarrow G.$GetNodeFeatures ();
11  Working memory $\boldsymbol{M}_w \leftarrow$ GATv2($\{\boldsymbol{V}, n_{global}\}$);
12  $\boldsymbol{e}_{cur} \leftarrow \mathcal{F}_{enc}(\mathbf{l}_t), \boldsymbol{e}_{target} \leftarrow \mathcal{F}_{enc}(\mathbf{l}_{target})$;
13  $f_{cur} \leftarrow \mathcal{D}_{cur}\,(e_{cur}, \boldsymbol{M}_w)\,,\ f_{target} \leftarrow \mathcal{D}_{target}(\boldsymbol{e}_{target}, \boldsymbol{M}_w)$;
14  $\alpha \leftarrow \mathcal{D}_{target}.$GetAttScores()
  // Step 4:  Action generation
15  $\boldsymbol{x} \leftarrow$ LSTM(FC($[\boldsymbol{f}_{cur}, \boldsymbol{f}_{target}, \boldsymbol{e}_{cur}]$));
16  $p\,(a_t \mid x) = \sigma(\text{FC}(x))$;
17  $a_t \leftarrow$ SampleFromDistribution($p(a_t \mid \boldsymbol{x})$);
18 **end**

---

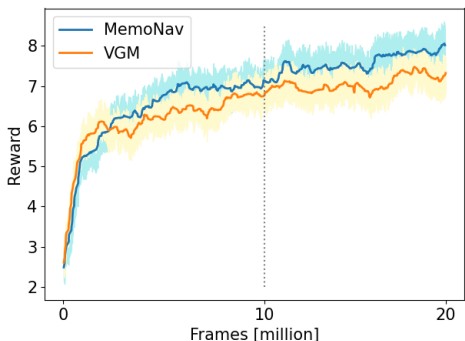

Figure 9: Reward curves of the MemoNav and VGM during the RL training phase.

### A.4.2   REPRODUCTION OF CNNLSTM

We reproduce CNNLSTM (Zhu et al., 2017) following the description in its original paper, but we also make some modifications to keep the comparison fair. We replace the ResNet-50 in CNNLSTM with the pretrained RGB-D encoder of VGM (Kwon et al., 2021). We also add positional embeddings to the encoded RGB-D observations to contain temporal information. Moreover, we concatenate the encoded RGB-D observations with the goal image embedding and project the concatenated feature (1024D) to a 512D feature, so that CNNLSTM can utilize the information of the goal image. The projected features of four consecutive frames are further condensed and then input to a policy network as in (Zhu et al., 2017). To use the two auxiliary tasks proposed in VGM (Kwon et al., 2021), we also introduce the linear projection layers (Linear-ReLU-Linear) used in VGM to process the embedded goal image and embedded current observation. With these modifications, the comparison between CNNLSTM and MemoNav (ours) is fair.

### A.4.3   TRAINING DETAILS

We follow the two-step training routine and maintain the training hyper-parameters in the VGM paper (Kwon et al., 2021). Firstly, CNNLSTM, VGM, and MemoNav (ours) are all trained using imitation learning for 20k steps. Afterward, we finetune these models using PPO (Schulman et al., 2017) for 10M steps. Due to the performance fluctuation intrinsic to reinforcement learning, the model at the 10M-th step is probably not the best. Therefore, we evaluate all checkpoints in the step range [9M, 10.4M] and select the best one.

A recent study (Yadav et al., 2022) has pointed out that training via PPO does not converge till 500M frames and results at 10M are highly sensitive to initialization. For better comparing the MemoNav and VGM, these two methods are trained for 10M more steps. The reward curves during the training phase is shown in Figure 9. This figure shows that the reward gain obtained by MemoNav becomes larger when the number of training steps increases from 10M to 20M.

### A.4.4   EXTRA INFORMATION ABOUT THE MULTI-GOAL DATASETS

We follow the format of the public 1-goal dataset in (Mezghani et al., 2021) and create 2-goal, 3goal, and 4-goal test datasets on the Gibson (Xia et al., 2018) scenes. We generate 50 samples for each of the 14 test scenes. In each sample, we randomly choose target positions while still following five rules: (1) no obstacles appear inside a circle with a radius of 0.75 meters centered at each target; (2) the distance between two successive targets is no more than 10 meters; (3) all targets are placed on the same layer without altitude differences. (4) all targets are reachable from each other. (5) The final target is placed near a certain previous one with the distance between them smaller than 1.5 meters. The distributions of the total geodesic distances for the three difficulty levels are shown in Figure 10.

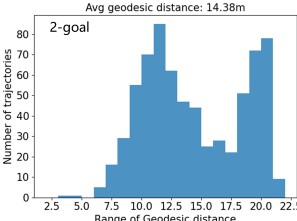 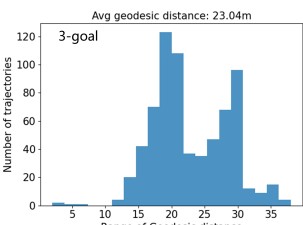 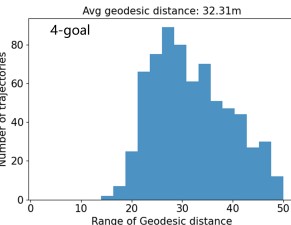

Figure 10: Histograms of geodesic distances for the multi-goal test datasets.

### A.4.5 COMPUTE REQUIREMENTS

We utilize an RTX TITAN GPU for training and evaluating our models. The imitation learning phase takes 1.5 days to train while the reinforcement learning takes 5 days.

The computation in the GATv2-based encoder and the two Transformer decoders occupy the largest proportion of the run-time of the MemoNav. The computation complexity of the encoder and the decoders are $\mathcal{O}(|\mathcal{V}|d^2 + |\mathcal{E}|d)$ and $\mathcal{O}(|\mathcal{V}|d)$, respectively. Using the forgetting module with a percentage threshold $p$, the computation complexity of the MemoNav can be flexibly decreased by reducing the number of nodes to $p|\mathcal{V}|$.

### A.5 EXTRA ABLATION STUDIES

We conduct additional ablation experiments on the forgetting module in the MemoNav to further investigate how its design affects the agent's navigation performance. The results are shown in Table 4.

**Origin of attention scores used by forgetting module**. The forgetting module in the MemoNav removes the uninformative STM according to the attention scores generated in $\mathcal{D}_{target}$, as described in Section 4.2.1. We change these scores to those generated in $\mathcal{D}_{cur}$. The result (row 2) shows that using the attention scores generated in $\mathcal{D}_{cur}$ leads to slightly worse performance.

We also remove STM according to the average of the scores generated in $\mathcal{D}_{cur}$ and those in $D_{target}$. The result (row 3) demonstrates that this variant obtains lower SR and SPL in the 1-goal setting but larger PR and PPL in harder tasks (i.e., 3 and 4-goal tasks). We hypothesize that this phenomenon occurs because averaging scores can include context information close to the agent, and not just the goal. Easy tasks require less memory, so including too many scene features around the agent leads to frequent changes in the output actions, thereby undermining the performance in 1 and 2-goal settings. In contrast, harder tasks require the agent to utilize its context information to plan a short path to visited areas, so retaining memory informative for both the current position and the goal benefits navigation performance.

**Effectiveness of forgetting module**. This ablation investigates whether it is effective to exclude the STM with attention scores ranking below the predefined percentage $p$. "Effectiveness" here means that the forgetting module does retain useful node features. In this experiment, the forgetting module excludes a random fraction of the STM. We test this ablation model over five random seeds and report the average metrics. The result (row 3) shows that incorporating a random fraction of the STM into the WM leads to decreases in all metrics, which validates the effectiveness of our design for retaining informative STM.

To better understand how the forgetting module works, we conduct an additional statistical experiment. In this experiment, five distance metrics are calculated: (a) distance from a node to the agent, (b) distance from a node to the goal, (c) distance from a node to the oracle shortest path, (d) distance from a node to the shortest path segments closer to the agent, and (e) distance from a node to the shortest path segments closer to the current goal. Then the histograms of these five metrics are drawn according to the metrics records for each forgotten/retained node at each time step so that we can see how far away these nodes are from the agent, the goal, and the shortest path. Please see Figure 11 to better understand the definitions of distance metrics (c)(d)(e).

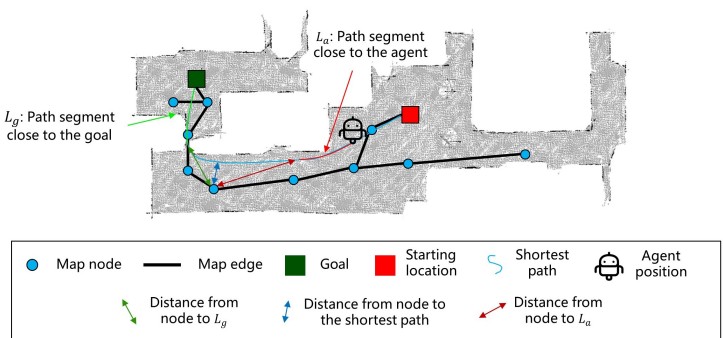

Figure 11: The visualization of distance metrics (c), (d), and (e).

Table 4: Extra ablation studies of the forgetting module.

| Methods | 1-goal | | 2-goal | | 3-goal | | 4-goal | |
|---|---|---|---|---|---|---|---|---|
| | SR | SPL | PR | PPL | PR | PPL | PR | PPL |
| 1 MemoNav | $60.7_{(2.1)}$ | $49.0_{(1.9)}$ | $47.0_{(2.4)}$ | $18.3_{(1.1)}$ | $35.8_{(1.4)}$ | $8.6_{(0.4)}$ | $27.5_{(1.0)}$ | $5.1_{(0.2)}$ |
| 2 w. $\mathcal{D}_{cur}$ att. scores | $60.4_{(1.4)}$ | $48.6_{(1.4)}$ | $46.9_{(1.8)}$ | $18.2_{(1.0)}$ | $35.7_{(1.5)}$ | $8.6_{(0.3)}$ | $28.4_{(1.4)}$ | $5.2_{(0.2)}$ |
| 3 w. averaged scores | $60.1_{(0.6)}$ | $48.4_{(0.6)}$ | $46.9_{(1.2)}$ | $18.4_{(0.6)}$ | $36.7_{(1.2)}$ | $8.8_{(0.4)}$ | $28.4_{(1.5)}$ | $5.4_{(0.2)}$ |
| 4 w. Random STM | $59.7_{(2.0)}$ | $44.6_{(1.7)}$ | $44.9_{(1.3)}$ | $15.3_{(0.9)}$ | $33.1_{(1.2)}$ | $8.1_{(0.4)}$ | $25.7_{(1.1)}$ | $4.8_{(0.2)}$ |

We evaluate the MemoNav on the 3-goal Gibson task and draw the histograms **on per-goal basis** with the average results of five different seeds, as shown in Figure 12. The figure provides two interesting findings:

- The distance distribution patterns for forgotten nodes (green) and retained ones (orange) vary across goals. The distributions of the distances from forgotten nodes to goals (column 2) and to shortest path segments near goal (column 5) become uniform when the agent navigates to the third goal. In contrast, these two histograms for the retained nodes become sharper and the peaks shift to smaller distance values. This phenomenon occurs because the forgetting module selectively retains map nodes closer to and informative for the current goal.

- The forgetting module has a larger impact on the distance metrics when the navigation task becomes more difficult. Specifically, when the current goal index is 1 (i.e. the task is easy), the averages of the distance metrics for forgotten nodes and retained nodes are close. When the goal index rises to 3 (i.e. the task becomes harder), a larger proportion of the retained nodes are close to the goal, the shortest path, and the shortest path segments near goal while a larger proportion of the forgotten nodes are close to the agent, and the shortest path segments near agent. This difference suggests that MemoNav improves navigation performance by focusing more on the region along the shortest path and around the goal area.

These results empirically validate that the information useful for path planning are not totally lost by the forgetting module.

**Extra Validation of LTM**. In order to observe how the learned scene-level representation in the LTM benefits the MemoNav, we replace the LTM feature with the average of all STM features. After replacing, the LTM no longer contains a scene-level representation, but it still facilitates message passing. The results in Table 5 shows that the MemoNav without the scene-level representation in the LTM witnesses drops in all metrics. This experiment suggests that the learned scene-level representation in the LTM contains important information that helps to improve navigation performance.

### A.6 EXTRA VISUALIZATION RESULTS

**More examples of multi-goal episodes** are displayed in Figure 13(a). The agent efficiently explores the scenes and finds sequential goals using the informative fraction of the node features in the map.

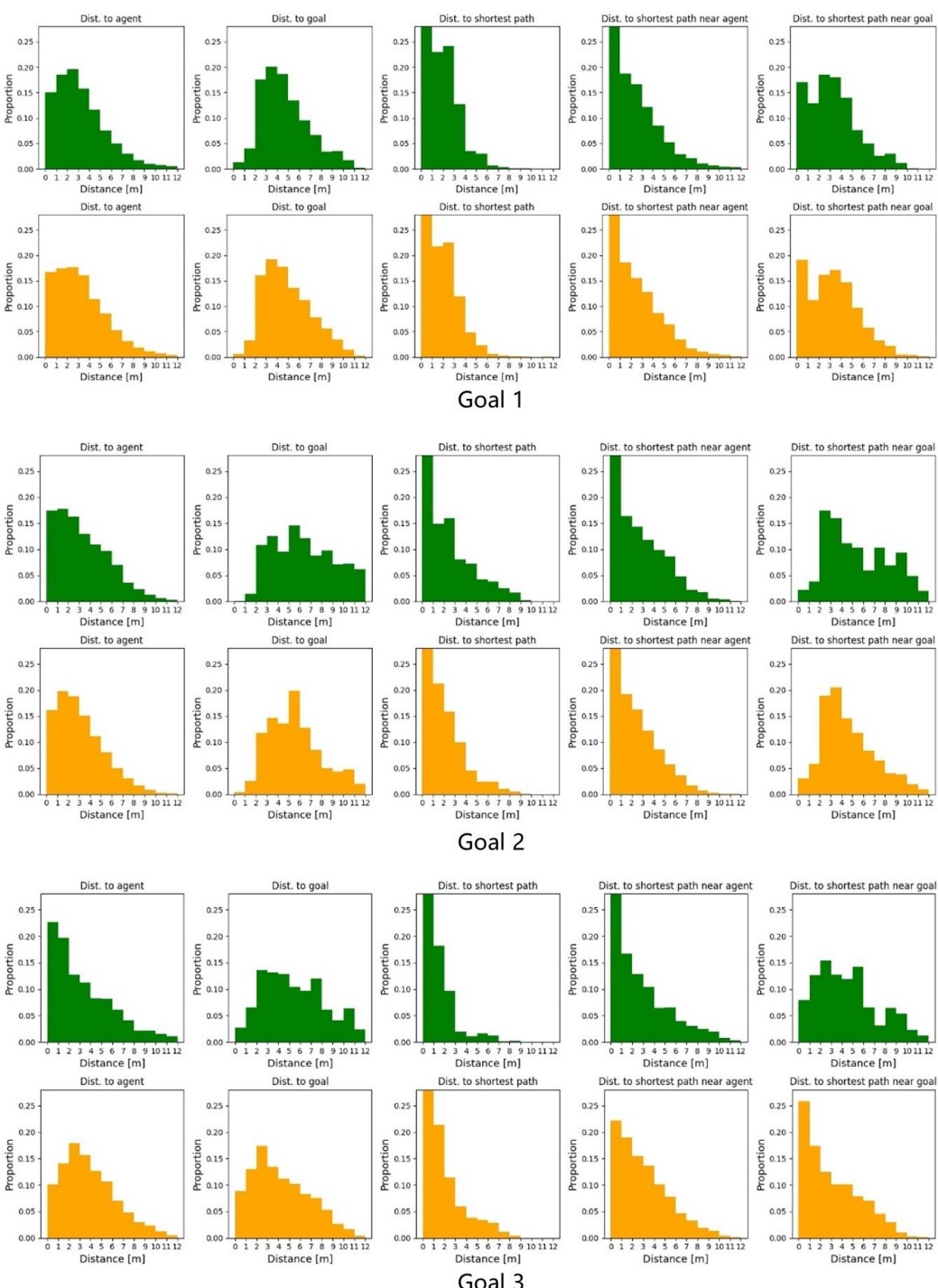

Figure 12: **Histograms of the five distance metrics defined in Section A.5**. The data of these metrics is collected by evaluating the MemoNav on the 3-goal task in the Gibson scenes and averaged over five runs. The upper row (green) and lower row (orange) belong to the forgotten nodes and retained ones, respectively.

Table 5: Extra ablation studies of the LTM.

| | Methods | 1-goal | | 2-goal | | 3-goal | | 4-goal | |
|---|---|---|---|---|---|---|---|---|---|
| | | SR | SPL | PR | PPL | PR | PPL | PR | PPL |
| 1 | MemoNav | $60.7_{(2.1)}$ | $49.0_{(1.9)}$ | $47.0_{(2.4)}$ | $18.3_{(1.1)}$ | $35.8_{(1.4)}$ | $8.6_{(0.4)}$ | $27.5_{(1.0)}$ | $5.1_{(0.2)}$ |
| 2 | w. averaging STM as LTM | $58.1_{(1.6)}$ | $47.2_{(1.5)}$ | $45.1_{(1.6)}$ | $17.6_{(0.8)}$ | $34.4_{(1.3)}$ | $8.1_{(0.3)}$ | $26.4_{(0.8)}$ | $4.5_{(0.2)}$ |

These examples show that the MemoNav agent focuses high attention only on a small fraction of nodes and excludes nodes that are far away from the current goal. For example, in the 3-goal example, the agent forgets the topmost node when navigating to the 1st goal since this node is the farthest from the goal; the agent forgets the nodes at the bottom left corner when navigating to the 2nd and 3rd goals since these nodes are remote and uninformative. The comparison with the baseline for these examples is recorded in the supplementary videos.

**Failure case analysis**. We present examples of failed episodes in Figure 13(b) and record the proportions of various failure modes at all difficulty levels. The failure modes can mainly be categorized into four types: *Stopping mistakenly*, *Missing the goal*, *Not close enough*, and *Over-exploring*. The mode *Stopping mistakenly* means that the agent implements stop at the wrong place. The mode *Missing the goal* means that the agent has observed the goal but passes it. The mode *Not close enough* means that the agent attempts to reach the goal it sees but implements stop outside the successful range. The mode *Over-exploring* means that the agent spends too much time exploring open areas without any goals. The largest probability lies in *Over-exploring* cases, most of which occur when the agent explores a large proportion of the scene but still fails to get close to the target area in a limited time.

**Visualization comparison between MemoNav with and without forgetting.** We compare the trajectories of the MemoNav with and without the forgetting module to observe how this module affects the characteristics (e.g. smoothness and length) of the trajectories. Figure 14 shows that the trajectories of MemoNav using the forgetting module are smoother and shorter. Without the forgetting module, the trajectories tends to possess many tiny sharp turns and are longer. We hypothesize that a fraction of the STM is uninformative for navigation. This fraction may acts as noise and make the policy network frequently change its output actions. The forgetting module adaptively excludes this disturbing fraction of STM so that the policy network is able to use helpful navigation memory for decision-making.

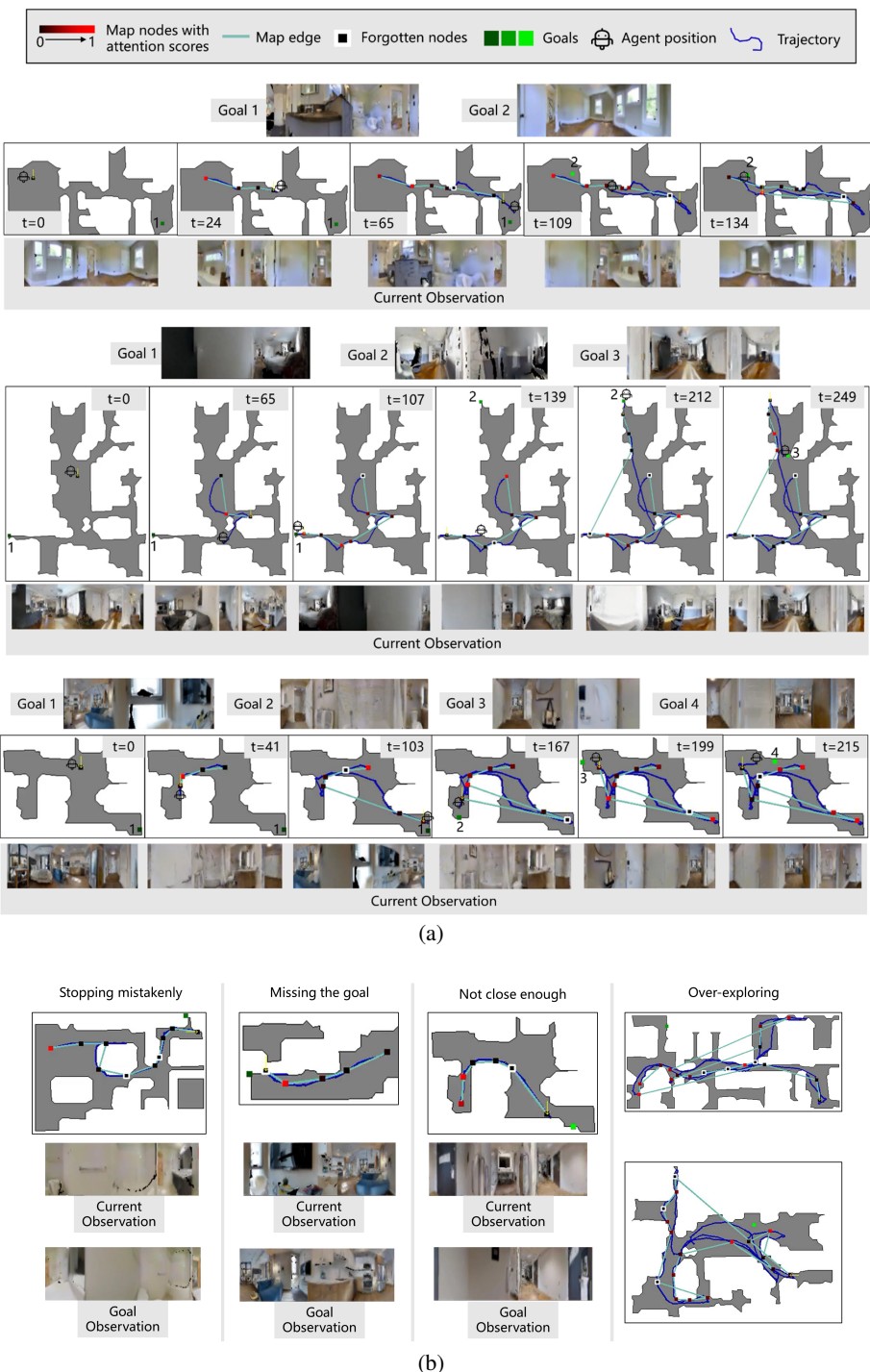

Figure 13: (a) Examples of multi-goal navigation episodes. Each example shows both the topological map and the trajectory. The graph nodes are incrementally added to the map by the agent and selectively retained by the forgetting module in the MemoNav. The yellow downward arrow denotes the current localized node of the agent. (b) Examples of failed episodes. The agent encounters four major failure mode: *Stopping mistakenly*, *Missing the goal*, *Not close enough*, and *Over-exploring*.

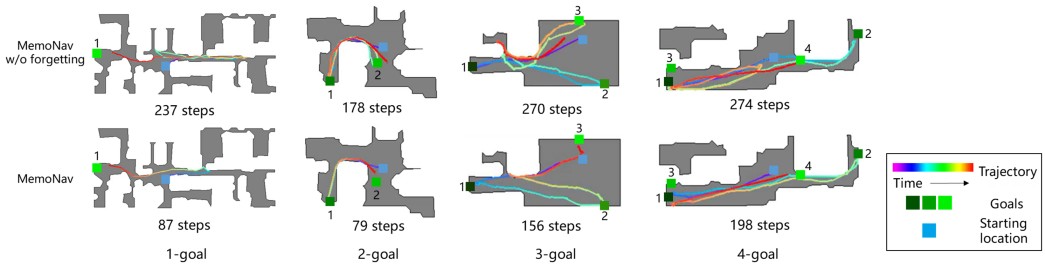

Figure 14: **Visualization comparing the MemoNav with and without the forgetting module.** We compare selected episodes at four difficulty levels in the Gibson scenes and visualize the top-down views. The number of navigation steps (the upper limit is 500) are shown at the bottom of each top-down view. Best viewed in color.

