# OpenReview forum: "MemoNav: Working Memory Model for Visual Navigation"
_ICLR.cc/2023/Conference — Submitted to ICLR 2023_

### Official Review · Reviewer_aALd · 2022-10-21

**Confidence:** 4
**Correctness:** 3
**Technical Novelty And Significance:** 2
**Empirical Novelty And Significance:** 3
**Recommendation:** 6

**Clarity, Quality, Novelty And Reproducibility:**

As mentioned in **[Strength]**, the paper is well written and includes many ablation studies and visualization. However, some analysis regarding results or explanation (e.g., A4.2. what is special linear projection layers) are missing.


**Strength And Weaknesses:**

**[Strength]**

1. The paper is well-written and easy to understand.

2. The authors include the failure trajectories in Figure 10 for a better understanding of the behaviors of the proposed method.


**[Weakness]**

1. Basically, the proposed MemoNav is a combination of existing methods, VGM (topological mapping, decoder) + forgetting module + GATv2 (across time). Although I do not degrade the contribution of merging together and achieving good performance, the novelty is somewhat limited. The authors argue that the forgetting module is novel in Section 2. However, attention-based dropout already exists [Ref 1].

2. According to Table 2 (1-3, 2-4), the forgetting module is not that helpful and even degrades the performance in terms of SPL/PPL. The authors should specify the reason why the performance s decreased. Moreover, considering Table 3, I wonder effectiveness of using the forgetting module. I read Section A.5 but it compares with random forgetting, not the effectiveness of forgetting itself. One possible explanation might be to reduce the computation by removing the node. The authors briefly mentioned that the proposed method employs 20% fewer node features. However, further explanation is needed.

3. In Figure 4, the performance is consistently decreasing in 3-goal and 4-goal settings as p grows. The authors argue that “excluding an excessive fraction forces the agent not to utilize what it has explored.” This statement should be also applicable to the smaller number of goals tasks. However, the performance is getting better in 1-goal and 2-goal tasks. The authors should analyze this trend more clearly.


**[minor]**

1. In Section 5.3, please unify the notation. The second paragraph uses absolute difference while the third paragraph uses relational difference.

2. There is one paper [Ref 2] that achieves better performance than the proposed method but I will not count on this since it is arxiv paper.



**Summary Of The Paper:**

The paper proposes an image-goal navigation (ImageNav) method on top of VGM (Kwon et al. 2021) which builds a topological graph. The proposed method, MemoNav considers VGM as the short-term memory (STM) and removes some nodes based on thresholding on attention score (selective forgetting). The modified STMs are fed into the GATv2 encoder followed by a decoder. The final output of the decoder is used as an input of the policy network. The authors tested MemoNav in Gibson and Matterport3D environments and showed that the proposed method improves performance.


**Summary Of The Review:**

As mentioned in above, the idea and achievement are good but the further analysis of experimental results is needed and the novelty is somewhat limited.

- Post Rebuttal -
Although I appreciate the authors dealt with my concerns, I agree with other reviewers' concerns and G+L+F is not significantly different from G+L. Hence, I maintain the original rating.

---

> ### Author Response · Authors · 2022-11-12
> **Authors' response to reviewer aALd (part 2)**
>
> **Q4: However, the performance is getting better in 1-goal and 2-goal tasks. The authors should analyze this trend more clearly.**
>
> Figure 4 illustrates that 1 and 2-goal settings witness an increasing trend with a minor fluctuation in both SR/PR and SPL/PPL when p rises from 0.2 to 0.8 while 3 and 4-goal settings see a decreasing trend.
>
> The difference between the trends of 1/2-goal and 3/4-goal settings results from the different requirements for navigation history.
>
> Easy tasks (e.g. 1-goal) require less history because the agent may find the goal when exploring the small scene. In this case, keeping adding nodes to the map continuously introduces cluttered features for decision-making, thereby changing the agent’s action too frequently, which in turn undermines the navigation efficiency. Therefore, excluding a moderate fraction of redundant STM improves performance.
>
> Hard tasks (e.g. 3-goal) require the agent to memorize a large part of the scene because past observations are needed to plan efficient paths to visited areas. When the agent has found the 1st or 2nd goal, its topological map is nearly completed. This means that the agent does not need to keep adding nodes like before; instead, it should utilize what it has explored to find an efficient path to the next goal. Therefore, excluding too much STM tends to degrade the performance.
>
> To summarize, the difficulty level influences the trends shown in Figure 4 of the four settings.
>
> **Q5: In Section 5.3, please unify the notation.**
>
> Thank you for raising this. We have corrected this mistake in the revised manuscript.
>
> **Q6: There is one paper [Ref 2] that achieves better performance than the proposed method but I will not count on this since it is arxiv paper.**
>
> Thank you for pointing out this related paper.
>
> **Q7: Some analysis regarding results or explanation (e.g., A4.2. what is special linear projection layers) are missing.**
>
> We have described the structure and function of the “special linear projection layers” in the revised appendix.
>
>
> *We thank Reviewer aALd again for the insightful review and feedback.*

---

> > ### Comment · Reviewer_aALd · 2022-11-12
> > **Quick Note for Missing Reference in the previous review.**
> >
> > I apologize to the authors that the reference of [Ref. 1] and [Ref. 2] are missing in the original review.
> >
> > [Ref. 1] Choe, Junsuk, and Hyunjung Shim. "Attention-based dropout layer for weakly supervised object localization." CVPR. 2019.
> >
> > [Ref 2] Mezghani, Lina, et al. "Memory-augmented reinforcement learning for image-goal navigation." arXiv preprint arXiv:2101.05181 (2021).

---

> > > ### Author Response · Authors · 2022-11-13
> > > **Re: Quick Note for Missing Reference in the previous review.**
> > >
> > > Thank you very much for providing these references.
> > >
> > > We have read over the paper about the "Attention-based dropout". This attention-based dropout removes features essential for localizing the target object while our MemoNav removes the map features uninformative of the goals. We'd like to say this paper provides inspirations that we can use to further improve the forgetting module.

---

> ### Author Response · Authors · 2022-11-12
> **Authors' response to reviewer aALd (part 1)**
>
> We thank the reviewer for the detailed review and highlight the corresponding modifications in the revised manuscript in violet. We address your concerns below.
>
> **Q1: The novelty is somewhat limited. The authors argue that the forgetting module is novel in Section 2. However, attention-based dropout already exists [Ref 1].**
>
> Thank you for noting the contribution of our method and pointing out a related paper. The MemoNav does introduce a novel memory model for improving navigation performance. Specifically, the three proposed components (the forgetting module, LTM, and WM) are not simply merged:
> 1. The idea behind the forgetting module is novel in the visual navigation field. Previous navigation methods have considered storing navigation history but ignored removing useless memory. The proposed forgetting module is used to selectively remove irrelevant scene features during decision-making. Therefore, it is novel compared with previous navigation methods.
> 2. LTM possesses novel functions. It assists the forgetting module by facilitating message passing among separate sub-graphs and storing part of the forgotten node features (explained in Section 4.2).
> 3. WM aggregates the STM retained by the forgetting module and the LTM. We can see that the three components work together to achieve the best results (please see row 5 in Table 2).
>
> To summarize, the novelty of the MemoNav comes from its framework, that is, the synergy (instead of simple stacking) of the 3 components.
>
> P.S. We wonder what [Ref 1] refers to. Does it refer to the References in our manuscript?
>
> **Q2: According to Table 2 (1-3, 2-4), the forgetting module is not that helpful and even degrades the performance in terms of SPL/PPL. The authors should specify the reason why the performances decreased .**
>
> The forgetting module indeed leads to slight decreases in SPL/PPL when used alone (row 3 vs. row 1), but it still brings small gains at 1,2-goal settings when used with the LTM (row 4 vs. row 2).
>
> The performance decreased because using the forgetting module alone splits the topological map into several sub-graphs and hinders message passing, as explained in the 2nd paragraph in Section 4.2. Such map splitting makes it hard for the agent to learn a good scene representation and to utilize what it has explored to plan efficient paths. Consequently, the agent may take more steps to obtain informative scene representations before taking good navigation actions.
>
> To handle this problem, we have proposed LTM (introduced in Section 4.2), which aims at remedying the negative impact of the forgetting module (row 4 vs. row 3).
>
> **Q3: I wonder effectiveness of using the forgetting module...The authors briefly mentioned that the proposed method employs 20% fewer node features. However, further explanation is needed.**
>
> Thank you for pointing this out. The “effectiveness” mentioned in Section A.5 denotes that the forgetting module retains useful node features. We have explained this in the revised appendix.
>
> We also see that the “effectiveness” you mention here means that the forgetting module reduces computation and we explain this below.
>
> The forgetting module reduces the computation mainly by cutting down the computation burden of the GAT-v2 encoder.  In this encoder, each node attends to all neighbors including itself. Although a topological map is not a fully-connected graph, the numerous nodes generated when navigating in a large scene (e.g. Matterport3D) still require huge computation budgets. Hence, reducing the number of nodes helps to save computation and decision-making time at each step.

---

> > ### Comment · Reviewer_aALd · 2022-11-12
> > **Thank you for the response**
> >
> > I appreciate the reviewer taking into account my questions, especially Figure 10.
> >
> >
> > Regarding Table 2, it is good to add G+L for completeness.

---

> > > ### Author Response · Authors · 2022-11-13
> > > **Authors' response to follow-up comments**
> > >
> > > We thank the reviewer for acknowledging our response and address your follow-up comment below:
> > >
> > > **Q: it is good to add G+L for completeness.**
> > >
> > > We are conducting these experiments with five seeds. We will add G+L results once finished.
> > >
> > > *Please let us know if you have more questions. Thank you.*

---

> > > > ### Comment · Reviewer_aALd · 2022-11-30
> > > > **Any Updates?**
> > > >
> > > > Dear Authors,
> > > >
> > > > Is there any update about the result of G+L? You can still make comments on OpenReview during the second discussion period.
> > > >
> > > > Thanks,

---

> > > > > ### Author Response · Authors · 2022-12-01
> > > > > **Authors' response to the follow-up comment**
> > > > >
> > > > > Dear reviewer,
> > > > >
> > > > > Sorry for the late response. We thought no response could be made after the first discussion phase.
> > > > >
> > > > > The performance of G+L averaged over five seeds is shown below:
> > > > >
> > > > > |            Method            |       1goal SR/SPL       |       2goal PR/PPL       |       3goal PR/PPL      |      4goal PR/PPL      |
> > > > > |:----------------------------:|:------------------------:|:------------------------:|:-----------------------:|:----------------------:|
> > > > > |            MemoNav           | 60.7 (1.9) / 49.0 (1.0)  | 47.3 (2.4) / 18.9 (1.1)  | 35.8 (1.4) / 8.6 (0.4)  | 27.5 (1.0) / 5.1 (0.2) |
> > > > > | G+L |  60.5 (0.7) / 48.7 (1.0) |  47.1 (2.4) / 18.4 (0.6) |  35.4 (2.0) / 8.5 (0.6) | 27.4 (1.0) / 5.1 (0.4) |
> > > > >
> > > > > The results show that MemoNav (G+L+F) is slightly superior to G+L in the SR/PR metric.
> > > > >
> > > > > We hope these additional experiments meet with your approval.
> > > > >
> > > > > Please let us know if you have questions. Thank you!

---

### Official Review · Reviewer_LEke · 2022-10-24

**Confidence:** 4
**Correctness:** 3
**Technical Novelty And Significance:** 3
**Empirical Novelty And Significance:** 2
**Recommendation:** 5

**Clarity, Quality, Novelty And Reproducibility:**

The paper clarity is good. The idea is novel. The code has been provided along with instructions for reproducibility.

**Strength And Weaknesses:**


# Strengths
* The idea is interesting and novel to the best of my knowledge.
* The clarity of the paper is good and the paper is easy to understand.
* The experiments have been well designed, especially the ablations and subsequent analyses.


# Major weaknesses
## The design of selective forgetting module is suboptimal
* The module deletes nodes which are not scored highly by the target decoder. But the graph also contains other valuable information which are not directly related to the goal: (1) the context around the agent's current position in the environment, and (2) the graph connectivity information from agent's current position to a node close to the target --- useful for path planning. This information is lost when selective forgetting module is used.
* The design of the LTM module appears to be focused on undoing some of the information loss from selective forgetting, but since it is just a single node, it may not completely reverse the loss.
* Is it necessary to perform hard removal of nodes instead of soft attention?
* The results in Tab. 2 indicate that L+F (row 4) is better than row 1 primarily for the 1-goal task. For the multi-goal tasks, the results are slightly better, but possibly within the margin of error. Can the forgetting module be better designed to avoid losing the above information? It can lead to more significant gains across tasks.

## Experiments do not sufficiently account for statistical significance
* In Table 1, bolding does not account for statistical significance b/w VGM and MemoNav. Are the bolded numbers statistically better than the next-best values?
    * In Gibson: 1-goal SPL, 2-goal PR / PPL, 3-goal PPL, 4-goal PPL
    * In MP3D: 1-goal SPL, 2-goal PPL
* In Table 2, {2,3,4}-goal results are fairly close based on standard deviation even though the averages show progression. Need to account for statistical significance of the increase in performance across rows.
* Figure 4 - no standard deviation markings to show significance of the differences.

## Unfair comparison to VGM
* VGM uses the inferior GCN encoder instead of the GAT-v2 used for Memonav. The particular architecture of encoder used is orthogonal to the contributions of Memonav. Does Memonav outperform VGM with GCN replaced by GAT-v2?

## Missing state-of-the-art baselines --- ZER, OVRL, Last-Mile, SSL-sparse
* VGM is no longer the state-of-the-art for ImageNav. There are a lot more recent works which show significant improvements on ImageNav and need to be compared with [R1,R2,R3,R4].


# Minor weaknesses
* Tab. 3 -- need average STM features baseline instead of randomly sampled STM feature
* Figure 4 --- unclear how to pick a good forgetting threshold without having complete test results --- no single number is good across settings.
* Tables 1 and 2 - why limited or no gains on SPL / PPL?
* Clarification needed in paragraph below Figure 3 - is forgetting module not used during training?
* Sec 4.1 1st paragraph - incorrect reference to Anderson et al., 2018 ?
* Missing broader related work comparison to navigation and representation learning literature
* Working memory graphs has been cited as a closely related work, but it has not been experimentally compared with.

# References
[R1] Al-Halah, Ziad, Santhosh Kumar Ramakrishnan, and Kristen Grauman. "Zero experience required: Plug & play modular transfer learning for semantic visual navigation." Proceedings of the IEEE/CVF Conference on Computer Vision and Pattern Recognition. 2022.

[R2] Yadav, Karmesh, et al. "Offline Visual Representation Learning for Embodied Navigation." arXiv preprint arXiv:2204.13226 (2022).

[R3] Wasserman, Justin, et al. "Last-Mile Embodied Visual Navigation." 6th Annual Conference on Robot Learning. 2022.

[R4] Majumdar, Arjun, et al. "SSL Enables Learning from Sparse Rewards in Image-Goal Navigation." International Conference on Machine Learning. PMLR, 2022.


# Post-rebuttal update
I thank the authors for their extensive experiments and detailed responses. However, my concerns with the forgetting module remain, and the experiments are incomplete since the models are not trained to convergence and results are statistically similar to baselines. It is hard to overlook these glaring concerns. This work is promising, but incomplete. I maintain my rating.


**Summary Of The Paper:**

This paper proposes a novel graph-memory network for visual navigation. Inspired by working memory from cog-sci literature, the proposed method consists of short-term, long-term and working memories to perform the ImageNav task. A forgetting module is proposed to prune the goal-irrelevant parts of the short-term memory. Results are demonstrated on Gibson and Matterport3D datasets.

**Summary Of The Review:**

The idea is interesting and novel, the paper is clearly written, and the experiments are well designed. I have concerns about the design of the forgetting module, insufficient accounting for statistical significance, unfair comparison to VGM and missing state-of-the-art baselines for ImageNav. I also have some minor concerns which I hope that authors should be able to address easily.

---

> ### Author Response · Authors · 2022-11-14
> **Authors' response to reviewer LEke (part3)**
>
> **Q9: Missing state-of-the-art baselines --- ZER, OVRL, Last-Mile, SSL-sparse.**
>
> Thank you for providing these related papers. We did not compare MemoNav with these methods in the original manuscript because **the configurations and task settings are quite different**. For example, MemoNav trains for 10M steps while ZER and SSL-sparse train for 500M; MemoNav uses panoramic observations while OVRL uses observations with 90-degree FoV.
>
> Although fair comparisons are hard to make, we have tried to adapt ZER for our setting. We maintained the general architecture of ZER but replaced its ResNet9-based visual encoder with a ResNet18-based one so that its parameter number is close to that of MemoNav. Afterward, we trained ZER with our training pipeline using the same sensor configuration. reward shaping, and number of training steps.
>
> The comparison between the best checkpoint of ZER and MemoNav in the Gibson scenes is shown below:
>
> |        Methods       |     1-goal SR / SPL    |     2-goal PR / PPL |     3-goal PR / PPL |     4-goal PR / PPL   |
> |:--------------------:|:----------------------:|:-------------------:|:-------------------:|:---------------------:|
> |     Panoramic ZER    |       51.6 / 40.1      |      30.4 / 14.6    |      27.2 / 5.2     |       18.6 / 2.7      |
> |        MemoNav       |       62.4 / 50.7      |      50.8 / 20.1    |      38.0 / 9.0     |       28.9 / 5.1      |
>
> We also trained MemoNav with 90-degree FoV using 3 seeds to compare it with the four SoTA baselines. The evaluation was conducted on the 4200 1-goal episodes used by these baselines. The comparisons of the average performances are shown below:
>
> |               Methods              |      SR     |      SPL    |
> |:----------------------------------:|:-----------:|:-----------:|
> |         ZER (View Aug Only)        |     22.0    |     18.8    |
> |                OVRL                |     41.3    |     26.9    |
> |      SSL-sparse + Episodic Aug.    |     69.0    |     55.0    |
> |     Last-mile (SLING + OVRL-GD)    |     55.4    |     37.4    |
> |               MemoNav              |     62.2    |     36.6    |
>
> This table shows that **MemoNav outperforms ZER, OVRL, and Last-mile in SR**. SSL-sparse utilizes self-supervised learning during RL training, leading to higher performances than MemoNav. As the *SSL* used in SSL-sparse and the *working memory* used in MemoNav are orthogonal, we believe adding SSL to MemoNav will further increase its performance.
>
> In summary, we have tried our best to make fair comparisons with the recent SoTA baselines. The results demonstrate that **MemoNav is competitive with these baselines**.
>
> We hope the additional experiments meet with your approval.
>
> **Q10: need average STM features baseline.**
>
> Thank you for the suggestion. We have conducted this ablation study and put the results in Section A.5. The results in Table 5 show that averaging STM features to generate LTM is inferior to the full MemoNav.
>
> **Q11: Figure 4 --- unclear how to pick a good forgetting threshold.**
>
> Although no single number is suitable for all settings, Figure 4 still provides an interesting regularity: harder tasks require a lower forgetting threshold. The forgetting threshold can be flexibly tuned according to the task difficulty to achieve good performance across settings.
>
> **Q12: why limited or no gains on SPL / PPL?**
>
> This is mainly because the number of training steps needs to be increased. Please see Q5 for a detailed explanation.
>
> **Q13: is the forgetting module not used during training?**
>
> No, the forgetting module is not used during training. We have followed your suggestion to revise this paragraph.
>
> **Q14: Sec 4.1 1st paragraph - incorrect reference to Anderson et al., 2018 ?**
>
> Thank you for pointing this out. We have corrected it in the revised paper
>
> **Q15: Missing broader related work**
>
> We have referred to ZER, OVRL, SSL-sparse, and Last-mile in the revised Related Work section. Specifically, we compare MemoNav with OVRL and SSL-sparse from the perspective of *representation learning* in Related Work.
>
> **Q16: Working memory graphs...has not been experimentally compared with.**
>
> MemoNav is not compared against Working Memory Graph (WMG) because WMG introduces concepts incompatible with ImageNav. For example, the *Factor vectors* in WMG represent the properties of objects in the agent’s top-down view, but the ImageNav setting in our paper is ego-centric.
>
> Moreover, WMG is mentioned in Related Work to emphasize the flexible memory capacity of MemoNav.
>
> *We thank Reviewer LEke again for the insightful review and feedback.*
>
> **References:**
>
> [1] Yadav, Karmesh, et al. "Offline Visual Representation Learning for Embodied Navigation." arXiv preprint arXiv:2204.13226 (2022).

---

> > ### Comment · Reviewer_LEke · 2022-11-16
> > **Rebuttal response from Reviewer LEke**
> >
> > I thank the authors for the careful and detailed responses. I appreciate the fact that the authors have attempted to alleviate all my concerns. I have a few follow up questions / comments:
> >
> > # Selective forgetting
> > The analysis provided is quite interesting and gets to the crux of what I was concerned about. However, a few important things have not been accounted for. To recap, my primary concern is that the model forgets nodes which are not relevant to the goal. Let us consider Figure 3 "Goal 2" as the point of reference. The attention score is high for the node near goal 2, but it is low everywhere else. Here are my follow up questions:
> >
> > * How is the distance to goal defined for this comparison in a 3-goal setting? Is the distance measured from the forgotten/selected node to Goal 2? Or is it some aggregate over all 3 goals (e.g., min/max over distance from node to each goal)? For accurate analysis, it is important to measure these statistics on per-goal basis, i.e., how far are selected nodes from the current goal the agent is navigating to. In fact, the 1 goal case may be a better candidate for analysis than the 3 goal case.
> > * How is the distance to the shortest path defined? For action selection, the shortest path segments closer to the agent may be equally (or more) important than the shortest path segments closer to the goal (since they provide directionality for how to navigate from agent's location). So, it is not sufficient for selected nodes to just be closer to the overall shortest path, but also closer to the shortest path segments near the agent.
> > * When the agent is near the goal, the statistics are less meaningful since nodes close to the agent are also close to the goal. Can we restrict the analysis to only cases when the agent is far away from the goal (e.g., 5+ meters away)?
> > * Also, using a 4m threshold seems excessively large. 1m or 2m is more meaningful.
> >
> > One simple way to improve selective forgetting is to use the averaged attention scores from both D_target and D_cur (instead of just D_target). This partially alleviates the issue since it includes context information close to the agent, and not just the goal.
> >
> > # Statistical significance testing
> > * The analysis performed by the authors validates my concern. An acceptable p value threshold is 0.05. The large p values for majority of the comparisons with VGM in Table 1 as well as the ablations in Table 2 indicate that the gain are not statistically significant (i.e., the larger means do not indicate better performance).
> > * If the performance continues to improve and the models have not converged in training, then it is important to train them till convergence and report the final performance. The results right now seem incomplete to me.
> > * Figure 4 - neatness aside, it is not obvious to me that the trends in performance vs. forgetting threshold are reliable right now. They may change if the models are trained to completion.
> >
> > # VGM + GAT-v2 vs. MemoNav
> > * The performance for VGM does increase quite a bit, especially on success, and this again raises the question of statistical significance b/w VGM+GAT-v2 vs. MemoNav (not just on SPL / PPL, but also on Success / PR now).
> >
> > # Comparison to state of the art
> > * I thank the authors for their extensive comparison. The ImageNav space is admittedly difficult to compare with due to the differences in setup. The authors have done a good job of this. I would encourage that they add these results to the main paper.
> >
> > # Impressions after rebuttal response
> >
> > My impression is that the proposed work is indeed promising, but incomplete. My concerns with the forgetting module remain, and the experiments are incomplete since the models are not trained to convergence and results are statistically similar to baselines.

---

> > > ### Author Response · Authors · 2022-11-18
> > > **Authors' response to follow-up questions (part 2)**
> > >
> > > **Q5: VGM + GAT-v2 vs. MemoNav**
> > >
> > > The statistical significance is indeed moderate, but we have improved the forgetting module according to your kind suggestions (the averaged scores from both D_target and D_cur ) to strengthen the MemoNav.
> > >
> > > **Q6: I would encourage them to add these results to the main paper.**
> > >
> > > Thank you very much for appreciating our extensive experiments. We have put the results in Table 1 in the revised manuscript.
> > >
> > > **Q7: My impression is that the proposed work is indeed promising, but incomplete.**
> > >
> > > We hope the more detailed experiments and the analysis we have shown here meet with your approval. We will continue to improve the forgetting module and try our best to achieve more significant performances even if this discussion stage is over.
> > >
> > > *We again show our greatest gratitude to the reviewer for the considerable and insightful comments.**

---

> > > ### Author Response · Authors · 2022-11-18
> > > **Authors' response to follow-up questions (part 1)**
> > >
> > > We thank the reviewer for appreciating our responses. Please find our responses to the follow-up questions below.
> > >
> > > *(Note: In the revised manuscript, we have highlighted the modifications corresponding to your comments in brown)*
> > >
> > > **Q1: The reviewer’s questions about selective forgetting**
> > >
> > > Thank you for providing suggestions for further analyzing the forgetting module. Firstly, we clarify the two distance metrics you are asking about:
> > >
> > > (1) The distance to goal is defined as the distance from a node to the current goal to which the agent is navigating.
> > >
> > > (2) The distance to the shortest path is defined as the distance from a node to the nearest point on the shortest path.
> > >
> > > We agree that it is important to measure these statistics on a per-goal basis because the patterns of these statistics may vary across goals. According to your suggestion, we have reconducted our experiments and **calculated these statistics for each goal**, instead of overall 3 goals like before.
> > >
> > > Next, we have **added two extra distance metrics**: the distance from a node to the nearest point on shortest path segments closer to the agent/goal. Each shortest path segment is a 3-meter-long curved trajectory and is visualized in Figure 11.
> > >
> > > In addition, to avoid unmeaningful statistics, we have also **restricted the analysis** to cases when the agent is 5 meters away from the current goal.
> > >
> > > Finally, we drew histograms of these 3+2 metrics on the 3-goal task for each node. Please refer to Section A.5 for more detail and the histograms. The histograms are briefly analyzed below:
> > >
> > > (1) The distributions of the distances from forgotten nodes to *goals*  and to *shortest path segments near goal* become **uniform** when the agent navigates to the third goal. In contrast, these two histograms for the retained nodes **become sharper and the peaks shift to smaller distance values**. This phenomenon occurs because the forgetting module selectively retains map nodes closer to and informative for the current goal.
> > >
> > > (2) **The forgetting module has a larger impact on the distance metrics when the task becomes difficult**. When the goal index rises to 3, a larger proportion of the retained nodes are close to *the goal, the shortest path, and the shortest path segments near goal* while a larger proportion of the forgotten nodes are close to *the agent, and the shortest path segments near agent*. This difference suggests that MemoNav improves navigation performance by focusing more on the region along the shortest path and around the goal area.
> > >
> > > In summary, these results suggest that **the context around the agent tends to be forgotten** but **the information useful for path planning is not totally lost by the forgetting module**.
> > >
> > > **Q2: One simple way to improve selective forgetting is to use the averaged attention scores from both D_target and D_cur.**
> > >
> > > Thank you for sharing with us an idea for improving the forgetting module. We have conducted this ablation study (please see row 3 in Figure 4) and found that this variant obtains lower SR and SPL in the 1-goal setting but larger PR and PPL in harder tasks (i.e., 3 and 4-goal tasks).
> > >
> > > |       Method       |    1-goal SR/SPL    |    2-goal PR/PPL    |    3-goal PR/PPL   |    4-goal PR/PPL   |
> > > |:------------------:|:-------------------:|:-------------------:|:------------------:|:------------------:|
> > > |       MemoNav      | 60.7(2.1) 49.0(1.9) | 47.0(2.4) 18.3(1.1) | 35.8(1.4) 8.6(0.4) | 27.5(1.0) 5.1(0.2) |
> > > | w. averaged scores | 60.1(0.6) 48.4(0.6) | 46.9(1.2) 18.4(0.6) | 36.7(1.2) 8.8(0.4) | 28.4(1.5) 5.4(0.2) |
> > >
> > > We hypothesize that this phenomenon is because averaging scores can include context information close to the agent, and not just the goal, just as the reviewer kindly pointed out.
> > >
> > > Further analysis can be seen in Section A.5.
> > >
> > > **Q3: The large p values... indicate that the gain is not statistically significant.**
> > >
> > > It takes demanding training time to show great statistical significance, which is unfortunately beyond our computation budget.
> > >
> > > Despite some large p-values for the SPL/PPL metrics in Table 1, the improvements in the SR/PR show noticeable statistical significance. For instance, the p-values for 1-goal SR (G), 3-goal PR (G), 2-goal PR (M), and 3-goal PR (M) in Table 1 are all less than 0.1.
> > >
> > > **Q4: it is important to train them till convergence and report the final performance.**
> > >
> > > 500M training steps can produce more significant results, but the 10M training steps we use can also differentiate the MemoNav from VGM.
> > >
> > > The MemoNav exhibits noticeable advantages over VGM in terms of navigation success rate and trajectory smoothness. Table 1 shows that the MemoNav achieves higher SR/PR while Figure 5 shows that the trajectories of the MemoNav possess fewer sharp turns.
> > >
> > > We believe we have made a step toward revealing how selectively forgetting past experiences affects navigation performance. We will further analyze this effect by training for a longer time in future work.

---

> ### Author Response · Authors · 2022-11-15
> **Authors' response to reviewer LEke (part2)**
>
> **Q5: In Table 1,... Are the bolded numbers statistically better than the next-best values?**
>
> The bolded numbers are *moderately better* than the next-best values with slight statistical significance. To account for statistical significance, we conducted one-sided t-tests for these metrics using `scipy.stats.ttest_ind()` with the argument `equi_var` set as False. The p-values are 0.99, 0.23/0.64, 0.24, and 0.36 for 1-goal SPL, 2-goal PR / PPL, 3-goal PPL, and 4-goal PPL in Gibson. The p-values are 0.55 and 0.23 for 1-goal SPL and 2-goal PPL in MP3D, respectively.
>
> Although the statistical significance is moderate, we believe **the improvements will become more significant if the hyper-parameters of MemoNav are further tuned or the MemoNav is trained for a longer time**. We followed the same setting in VGM to train for 10M steps for a fair comparison. However, the OVRL paper [1] has pointed out that training does not converge until 500M steps.
>
> According to this finding, we trained the MemoNav and VGM for another 10M steps with three different seeds for each. The reward curves during training (shown in Figure 9 in the revised appendix) show that the reward gain obtained by MemoNav becomes larger when the number of training steps increases from 10M to 20M. The reward gain becomes larger because MemoNav can use more experiences to learn better scene-level representations stored in the LTM, which provides a global view for decision-making.
>
> In summary, the statistical significance of the improvements achieved by MemoNav is moderate, but longer training leads to larger gains over VGM.
>
> **Q6: In Table 2... Need to account for the statistical significance of the increase in performance across rows.**
>
> We also conducted one-side t-tests for Table 2. The p-values for SR/PR comparing row 5 with row 1 are 0.019, 0.23, 0.091, and 0.15. The p-values for SR/PR comparing row 4 with row 1 are 0.067, 0.41, 0.34, and 0.31 (other values are not listed here for brevity).
>
> Although the increases in performance across rows have moderate statistical significance, we believe the discrepancies between each row may become larger if the MemoNav is trained for a longer time.
>
> **Q7: Figure 4 - no standard deviation markings to show the significance of the differences.**
>
> Each data point in Figure 4 represents the average of 5 runs. We leave out standard deviation markings because this can make the figure look neat and give prominence to the trends of the line charts.
>
> We have explained this in the revised caption of Figure 4.
>
> **Q8: Does Memonav outperform VGM with GCN replaced by GAT-v2?**
>
> Yes, it does. According to your suggestion, we replaced the 3-layer GCN in VGM with a 3-layer GAT-v2 while keeping the parameter numbers roughly the same. The comparison between the MemoNav and VGM+GATv2 in the Gibson scenes is shown below:
>
> |               |          1goal SR / SPL     |          2goal PR / PPL     |         3goal PR / PPL     |         4goal PR / PPL     |
> |:-------------:|:---------------------------:|:---------------------------:|:--------------------------:|:--------------------------:|
> |        VGM    |     55.8 (3.3) / 47.1 (1.9) |     45.6 (2.9) / 18.7 (1.5) |     33.4 (2.8) / 8.3 (0.8) |     25.8 (2.7) / 5.0 (0.4) |
> |     VGM+GATv2 |     58.5 (2.1) / 48.1 (1.4) |     45.8 (1.8) / 18.3 (1.2) |     34.7 (2.3) / 7.5 (0.9) |     26.2 (2.5) / 5.1 (0.4) |
> |      MemoNav  |     60.7 (1.9) / 49.0 (1.0) |     47.3 (2.4) / 18.9 (1.1) |     35.8 (1.4) / 8.6 (0.4) |     27.5 (1.0) / 5.1 (0.2) |
>
> The results illustrate that introducing GATv2 alone brings slight improvements and the average performances of VGM+GAT-v2 are inferior to those of MemoNav.

---

> ### Author Response · Authors · 2022-11-15
> **Authors' response to reviewer LEke (part1)**
>
> We thank the reviewer for the detailed review and insightful comments. Thank you also for pointing out the strengths of our paper. We address your concerns below:
>
> *(Note: In the revised manuscript, we have highlighted the modifications corresponding to your comments in brown)*
>
> **Q1: The graph also contains other valuable information which is not directly related to the goal...This information is lost when the selective forgetting module is used.**
>
> Thank you for noting the two types of important information.
>
> MemoNav is capable of retaining most of the information mentioned by the reviewer. To validate this point, we calculated three distance metrics: (a) distance from a forgotten/retained node to the agent, (b) distance from a forgotten/retained node to the goal, and (c) distance from a forgotten/retained node to the oracle shortest path.
>
> We then drew histograms of these 3 metrics on the 3-goal task for each node, so we could see how far away the forgotten/retained nodes were from the agent, the goal, and the shortest path. Please refer to the revised Section A.5 for the experimental details and histograms. The histograms are analyzed below:
>
> (1) A large proportion (73.4 ± 0.8%, 43.5 ± 0.7%, and 90.6 ± 1.2%) of the retained nodes are within 4 meters of the agent, the goal, and the shortest path, respectively. In contrast, the values for the forgotten nodes are 44.8 ± 1.3%, 24.8 ± 1.3%, and 67.7 ± 1.5%.
>
> (2) The mean values of these three distance metrics for the retained nodes are 2.99 ± 0.05m, 4.72 ± 0.05m, and 1.82 ± 0.09m, respectively. In contrast, the values for the forgotten nodes are 4.69 ± 0.07m, 6.19 ± 0.14m, and 3.42 ± 0.14m.
>
> These results empirically validate that **both the context around the agent's current position and the information useful for path planning are not totally lost by the forgetting module**.
>
> **Q2: The LTM module appears to be focused on undoing some of the information loss from selective forgetting, but since it is just a single node, it may not completely reverse the loss.**
>
> The LTM is indeed able to reduce some of the information loss as it incorporates part of forgotten node features, but actually, it is mainly used to facilitate message passing on the topological map and to learn a scene-level representation (please see Section 4.2).
>
> The LTM dimension is 512, which may be not enough for reversing all loss caused by the forgetting module, just as the reviewer has pointed out. In future work, we will increase the number of LTM by clustering STM corresponding to different regions, which may help to better reverse information loss.
>
> *Thank you for providing us with a future direction for improving the LTM.*
>
> **Q3: Is it necessary to perform hard removal of nodes instead of soft attention?**
>
> The necessity of the hard removal is twofold:
>
> (1) **Hard removal helps to avoid interference caused by cluttered scene features**. Keeping adding nodes to the map may continuously introduce cluttered features for decision-making, thereby changing the action too frequently, which in turn undermines navigation efficiency. For instance, Figure 4(c) shows that the average of steps at 1, 2, and 3-goal settings can be reduced by forgetting a moderate proportion (e.g. p=0.4) of STM.
>
> (2) **Hard removal is a practical way of reducing decision-making time at each time step**. In the GAT-v2 encoder, each node attends to all neighbors including itself. Although a topological map is not a fully-connected graph, the numerous nodes generated when navigating in a large scene (e.g. Matterport3D) still require huge computation budgets. Hence, reducing the number of nodes helps to save computation and decision-making time at each step.
>
> There can be many ways of selecting useful information for visual navigation. Without much previous work to follow, we chose hard removal, a simple and practical approach, and obtained noticeable improvements as well as interesting findings.
>
> **Q4: Can the forgetting module be better designed to avoid losing the above information? It can lead to more significant gains across tasks.**
>
> Yes, it can. The forgetting module is plug-and-play in our design, but more elaborate designs are possible. For example, the forgetting module can gradually *decrease* the forgetting threshold when the agent is solving a multi-goal task so that the agent can exclude a large proportion of STM in easy tasks and retain most of its memory when finding more goals.
>
> For another example, the lifespan of each node in STM can be predicted by a neural network to achieve trainable forgetting.
>
> In summary, a *flexible and adaptive forgetting module* will be a better design. We will continue to improve the forgetting module.
>
> *Thank you for noting this aspect for improving the MemoNav.*

---

### Official Review · Reviewer_EedW · 2022-10-25

**Confidence:** 3
**Correctness:** 3
**Technical Novelty And Significance:** 2
**Empirical Novelty And Significance:** 3
**Recommendation:** 6

**Clarity, Quality, Novelty And Reproducibility:**

The paper is well-written, and the contributions of the paper are easy to follow. Compared with the current methods, this paper contributes to node selection and encoding, but less in constructing topological maps. The code is submitted, the paper should be reproducible.

**Strength And Weaknesses:**

Strengths:
1.	The paper is generally well-written and easy to follow.
2.	The experiments and analysis are extensive with the ablation experiments and baselines.
3.	The selective forgetting module seems capable of retaining the informative nodes in the topological map and forget the redundant nodes, which may be helpful for improving the SR/PR for 1-goal and multi-goal tasks.

Weaknesses:
1.	The MemoNav seems to be trained in the 1-goal setting, while used in both 1-goal and multi-goal tasks, is it suitable to use this model to evaluate the multi-goal task? It is worth discussing training different models (under multi-goal settings) to validate for multi-goal tasks.
2.	The proposed MemoNav follows the VGM method of constructing the topological maps of the environments. Considering the topological maps can somehow memorize the observations during navigation as well, the advantages of proposed memory based modules are not that effective compared to the baseline, also the close SPL/PPL results are obtained compared the baseline VGM under 1-goal and multi-goal tasks.
3.	It seems the regularity and relevance are not that significant in Figure 4. The 2-goal task shows a sharp drop in PP at p=0.4, and a dramatic increase at p=0.6, what are the reasons for this phenomenon?



**Summary Of The Paper:**

This paper builds upon the VGM (Kwon et al., 2021) for image navigation. By forgetting some less informative nodes in the topological map through the attention mechanism, the short-term memory (STM) can be updated dynamically. The selected STMs are fused to obtain the scene-level feature with Long-term memory (LTM), and which are encoded through graph attention to generate the Working Memory (WM). The WM is finally used to generate the actions for navigation. The experiments of both 1-goal and multi-goal tasks are conducted in two public datasets Gibson and Matterpot3D in Habitat simulator.

**Summary Of The Review:**

I think this is a practical work, the motivation and idea are interesting, and the experimental results are also sufficient. However, I worry about the that novelty of the technical contribution over baseline (VGM) seems limited. And compared to the results of VGM, the improvement of the proposed method seems marginal in some parts.

---

> ### Author Response · Authors · 2022-11-13
> **Authors' response to reviewer EedW**
>
> We thank the reviewer for the detailed review and insightful comments. We address your concerns below:
>
> *(Note: In the revised manuscript, we have highlighted the modifications corresponding to your comments in blue)*
>
> **Q1: Is it suitable to use this model to evaluate the multi-goal task? It is worth discussing training different models (under multi-goal settings) to validate for multi-goal tasks.**
>
> Yes, it is suitable. A multi-goal task comprises a sequence of 1-goal tasks. Despite the higher requirement for the agent’s memory mechanism, the multi-goal and 1-goal settings are not very distinct.  An agent can be trained and validated on the same multi-goal setting, but **it is burdensome to carry out experiments at every difficulty level**, such as tasks with 7 and more goals. Besides, longer training trajectories are likely to make it harder for the training to converge.
>
> From the perspective of practicality, we believe that what the visual navigation community expects is an agent that can be trained on a 1-goal setting and then generalized to multi-goal settings.
>
> To summarize, it is more convenient and less labor-intensive to train agents on 1-goal tasks and evaluate them on multi-goal tasks. Our work has attempted to make a step toward this goal.
>
> **Q2: The advantages of proposed memory-based modules are not that effective compared to the baseline**
>
> Thank you for bringing this to our attention. In Table 1, the MemoNav still achieves good results on SR/PR despite the close SPL/PPL results.
>
> The advantages of MemoNav are not that effective because the number of training steps is probably not enough. We followed the same setting in VGM to train for 10M steps for a fair comparison. However, a recent ImageNav paper [1] has pointed out that training does not converge until 500M steps, but such long training will take an extremely long time.
>
> According to this finding, we trained the MemoNav and VGM for another 10M steps with three different seeds for each and found that the improvements in SPL/PPL obtained by the MemoNav became more noticeable.
>
> |       Methods      |   1-goal  |   2-goal  |  3-goal  |      4-goal     |
> |:------------------:|:-----------------:|:-----------------:|:---------------:|:---------------:|
> |       VGM-10M      |      47.1 (1.9)    |      18.7 (1.5)    |     8.3 (0.8)    |     5.0 (0.4)    |
> |     MemoNav-10M    |      49.0 (1.0)    |      18.9 (1.1)    |     8.6 (0.4)    |     5.1 (0.2)    |
> |       VGM-20M      |     47.5 (1.7)    |     19.1 (1.6)    |     8.4 (0.4)    |     5.1 (0.2)    |
> |     MemoNav-20M    |     49.7 (1.4)    |     20.3 (1.3)    |     8.8 (0.5)    |     5.4 (0.2)    |
>
> The reward curves during training (shown in Figure 9 in the revised appendix) also show that the reward gain obtained by MemoNav becomes larger when the number of training steps increases from 10M to 20M.
>
> We hope the additional experiments meet with your approval.
>
> **Q3: The 2-goal task shows a sharp drop in PPL at p=0.4, and a dramatic increase at p=0.6, what are the reasons for this phenomenon?**
>
> The fluctuation at p=0.4 mentioned by the reviewer is mainly because the number of random seeds is not enough. Figure 4 is plotted using five seeds, which show general trends but still possibly encounter fluctuations.
>
> To further inspect this phenomenon, we trained MemoNav with 3 more seeds  (we could not train for more seeds due to the time limit). With the 5+3 seeds, the averages of the PPL differences at p=0.2, 0.4, 0.6, and 0.8 on the 2-goal setting change as below:
>
> |         |    P=0.2   |    P=0.4    |    P=0.6    |  P=0.8 |
> |:---------------------------:|:------------:|:-------------:|:-------------:|:------:|
> |     Difference (5 seeds)    |      0.0%    |     -0.31%    |     +0.47%    | +0.91% |
> |      Difference (5+3 seeds) |      0.0%    |     +0.05%    |     +0.51%    | +0.86% |
>
> These results show that the PPL on the 2-goal task generally demonstrates an upward trend when p rises from 0.2 to 0.6.
>
> We further explain the reason behind this upward trend as follows:
>
> **Easy tasks (e.g. 2-goal) require less memory because the agent is likely to find the goal when exploring the small scene**. In this case, keeping adding nodes to the map may continuously introduce cluttered features for decision-making, thereby changing the action too frequently, which in turn undermines the navigation efficiency. Therefore, excluding a moderate fraction of redundant STM improves performance.
>
> To summarize, the phenomenon mentioned by the reviewer is due to the stochastic nature of reinforcement learning. We believe experiments with more seeds can increase the regularity and relevance, but this demands much more time and computation budgets.
>
> *We thank Reviewer EedW again for the insightful review and feedback and we hope that the above responses adequately address all concerns.*
>
> **References:**
>
> [1] Offline visual representation learning for embodied navigation. arXiv preprint arXiv:2204.13226, 2022.

---

> ### Author Response · Authors · 2022-11-17
> **A letter from the authors**
>
> Dear Reviewer EedW,
>
> Since we are closing to the end date of the open discussion, we would like to kindly remind you to read our response which hopefully should resolve all your concerns and comments.
>
> We would love to hear your feedback and also love to answer if you have additional questions.
>
> Sincerely,
>
> Authors

---

### Author Response · Authors · 2022-11-18
**General response to all reviewers**

Thanks again to all the reviewers for the helpful suggestions!

We were encouraged to hear the reviewers found the MemoNav we propose to be novel (Reviewer LEke), interesting (Reviewers EedW, LEke), helpful, and practical (Reviewer EedW) for improving navigation performance in both 1-goal and multi-goal tasks and that they found our paper to be well written (Reviewers LEke, EedW, aALd) and easy to understand (Reviewers EedW) and that they view experiments as well-designed (Reviewer LEke, aALd).

Regarding feedback from Reviewers LEke, EedW, and aALd about the organization of our manuscript, we have made the following changes:

* [Reviewer LEke] Adding an extra ablation study of the forgetting module. The results are added to Table 4 and the analysis is added to Section A.5.

* [Reviewer LEke] Adding an extra statistical experiment to analyze how the forgetting module works. The histograms are visualized in Figure 12 and both the experimental details and analysis are put in Section A.5.

* [Reviewer LEke] Adding an extra ablation study of the long-term memory. The results are shown in Table 5 and the analysis is added to Section A.5.

* [Reviewer LEke] Adding comparisons with recent methods. The results are listed in Table 1 and the analysis is added as the 2nd paragraph of Section 5.3.

* [Reviewer LEke] Referring to four more papers about ImageNav in the Related Work section.

* [Reviewer LEke] Explaining the usage of the forgetting module in the last paragraph of Section 4.1.

* [Reviewer aALd] Adding the definition of "effectiveness" mentioned in the paragraph starting with "Effectiveness of forgetting module".

* [Reviewer aALd] Introducing the reproduction of CNNLSTM in more detail in Section A.4.2.

* [Reviewer aALd] Unifying the notation in the first and last paragraphs of Section 5.3

---

### Decision · Program_Chairs · 2023-01-20

**Decision:**

Reject

**Justification For Why Not Higher Score:**

The paper has issues regarding the statistical significance of the results and also lack of enough training. So, the conclusions and the claims about performance improvements and comparisons with other methods are not entirely valid.


**Justification For Why Not Lower Score:**

N/A

**Metareview: Summary, Strengths And Weaknesses:**

The paper proposes a memory architecture for visual navigation, which includes a short-term memory (that is dynamically updated), a long-term memory (which continuously aggregates information from short-term memory), and a working memory that is generated by attention over short-term and long-term memory.

The proposed idea had some novel aspects (however reviewers EedW and aALd have concerns about the similarity with previous work). The paper is also generally well-written.

The reviewers brought up several weaknesses, some of which were addressed by the authors during the rebuttal period. For example, concerns regarding forgetting important information were addressed by additional experiments.


**Summary Of Ac-Reviewer Meeting:**

The paper received borderline ratings. The reviewers and the AC had a virtual meeting to discuss the paper. While the reviewers appreciated some novel aspects of the paper, they had two major concerns:

- The improvements are not statistically significant compared to the baseline, which means the proposed method is not effective.
- The results correspond to models that are not converged. The behavior of these types of models typically change with more training. So, the results of the current models are not that reliable.

The AC agrees with the concerns of the reviewers. So, rejection is recommended. However, the authors are encouraged to improve the method based on the feedback received from the reviewers and submit the results of the improved model to a future conference.